# A dual mechanism promotes switching of the Stormorken STIM1 R304W mutant into the activated state

Marc Fahrner[1], Michael Stadlbauer[1], Martin Muik[1], Petr Rathner[2], Peter Stathopulos[3], Mitsu Ikura[4,5], Norbert Müller [2,6] & Christoph Romanin [1]

STIM1 and Orai1 are key components of the $Ca^{2+}$-release activated $Ca^{2+}$ (CRAC) current. Orai1, which represents the subunit forming the CRAC channel complex, is activated by the ER resident $Ca^{2+}$ sensor STIM1. The genetically inherited Stormorken syndrome disease has been associated with the STIM1 single point R304W mutant. The resulting constitutive activation of Orai1 mainly involves the CRAC-activating domain CAD/SOAR of STIM1, the exposure of which is regulated by the molecular interplay between three cytosolic STIM1 coiled-coil (CC) domains. Here we present a dual mechanism by which STIM1 R304W attains the pathophysiological, constitutive activity eliciting the Stormorken syndrome. The R304W mutation induces a helical elongation within the CC1 domain, which together with an increased CC1 homomerization, destabilize the resting state of STIM1. This culminates, even in the absence of store depletion, in structural extension and CAD/SOAR exposure of STIM1 R304W leading to constitutive CRAC channel activation and Stormorken disease.

[1] Institute of Biophysics, Johannes Kepler University Linz, Gruberstrasse 40, 4020 Linz, Austria. [2] Institute of Organic Chemistry, Johannes Kepler University Linz, Altenbergerstrasse 69, 4040 Linz, Austria. [3] Department of Physiology and Pharmacology, University of Western Ontario, London, ON N6A 5C1, Canada. [4] Princess Margaret Cancer Center, University Health Network, Toronto, ON M5G 1L7, Canada. [5] Department of Medical Biophysics University of Toronto, Toronto, ON M5G 1L7, Canada. [6] Faculty of Science University of South Bohemia, Branišovská 1645/31A, 370 05 České Budějovice, Czech Republic. Marc Fahrner, Michael Stadlbauer, and Martin Muik contributed equally to this work. Correspondence and requests for materials should be addressed to M.F. (email: marc.fahrner@jku.at) or to C.R. (email: christoph.romanin@jku.at)

Store-operated calcium entry (SOCE) is essential for many calcium-mediated signaling pathways on the cellular and physiological level. Extracellular ligands bind to plasma membrane (PM)-resident proteins, which transduce the signal to the cell interior, triggering the release of calcium ($Ca^{2+}$) from the endoplasmic reticulum (ER), a process called store depletion[1]. The ER-located stromal interaction molecule1 (STIM1) senses the decrease in $[Ca^{2+}]_{ER}$ leading to STIM1 oligomerization and translocation to ER–PM junctions where it couples to and activates the $Ca^{2+}$-selective channel Orai1[2–11]. The significance of SOCE is highlighted by diverse mutations in STIM1 and Orai1 genes, which result in $Ca^{2+}$-release activated $Ca^{2+}$(CRAC) channelopathies that are characterized by autoimmunity, immunodeficiency, skeletal myopathy, and ectodermal dysplasia[6,12–18]. Recently, the human gain-of-function (GoF) mutant STIM1 R304W has been discovered in patients suffering of the Stormorken syndrome, which includes the symptoms thrombocytopenia, muscle fatigue, asplenia, miosis, migraine, dyslexia, and ichthyosis[18–22]. The cytosolic part of STIM1 interacts via direct coupling with both Orai1 C terminus and N terminus; however, the Orai1 C terminus has proven to be the dominant STIM1 coupling partner[23–27]. ER $Ca^{2+}$ store depletion is the initial trigger for STIM1/Orai1 interaction involving mechanistic steps, which include conformational changes of STIM1 resulting in oligomerization and translocation of STIM1 to the cell periphery[28,29]. The ER luminal-located STIM1 EF hand/SAM domain senses the decrease of $[Ca^{2+}]_{ER}$ and responds with oligomerization of the luminal STIM1 parts[3,4,30]. Consequently, the crossing transmembrane (TM) domains of STIM1 dimers congregate and change their angle resulting in conformational rearrangements of STIM1 cytosolic portions which finally lead to

oligomerization and extension of the respective domains[31]. The STIM Orai-activating region (SOAR aa344–442) or CRAC-activating domain (CAD aa342–448) represent ~100 amino acids within the STIM1 C-terminal strand responsible for coupling to and activating Orai1[27,32]. SOAR has been crystallized, revealing CC2 and CC3, forming an intramolecular antiparallel coiled coil, respectively[33]. Within full-length STIM1, the SOAR/CAD domain is kept quiescent in a tightly packed structure when ER stores are full. Upon store depletion, STIM1 extends its cytosolic strand, exposing SOAR/CAD for interaction with Orai[33–38]. CC1, which is upstream of SOAR/CAD, has an inhibitory role, as CC1α1 as well as CC1α3 exhibit an intramolecular interaction with CC3 resulting in a tight, quiescent STIM1 conformation[31,33,34]. CAD/SOAR exposure is triggered upon ER store depletion, releasing the CC1–CC3 clamp, therefore switching the cytosolic portion of STIM1 from a tight into an extended conformation[31,34–37]. CC2, which is part of CAD/SOAR, subsequently couples to the Orai1 C terminus, finally forming the STIM1-Orai1 association pocket as revealed by nuclear magnetic resonance (NMR)[38,39].

Recently, we have developed a live-cell imaging approach called FIRE (FRET interaction in a restricted environment), allowing us to dissect and analyze the mechanistic steps in the process of STIM1 activation[34]. We have precisely shown the contributions of homo-/heteromeric interactions of CC1α1, α2, and α3, as well as CC2 and CC3, allowing us to present a STIM1 conformational model integrating the tight and extended STIM1 state. In the present study, we explore the specific role of the GoF substitution R304W by employing electrophysiology, FRET (using the double-labelled Orai1-activating small fragment (OASF) conformational sensor and the FIRE system) and in vitro biochemical methods.

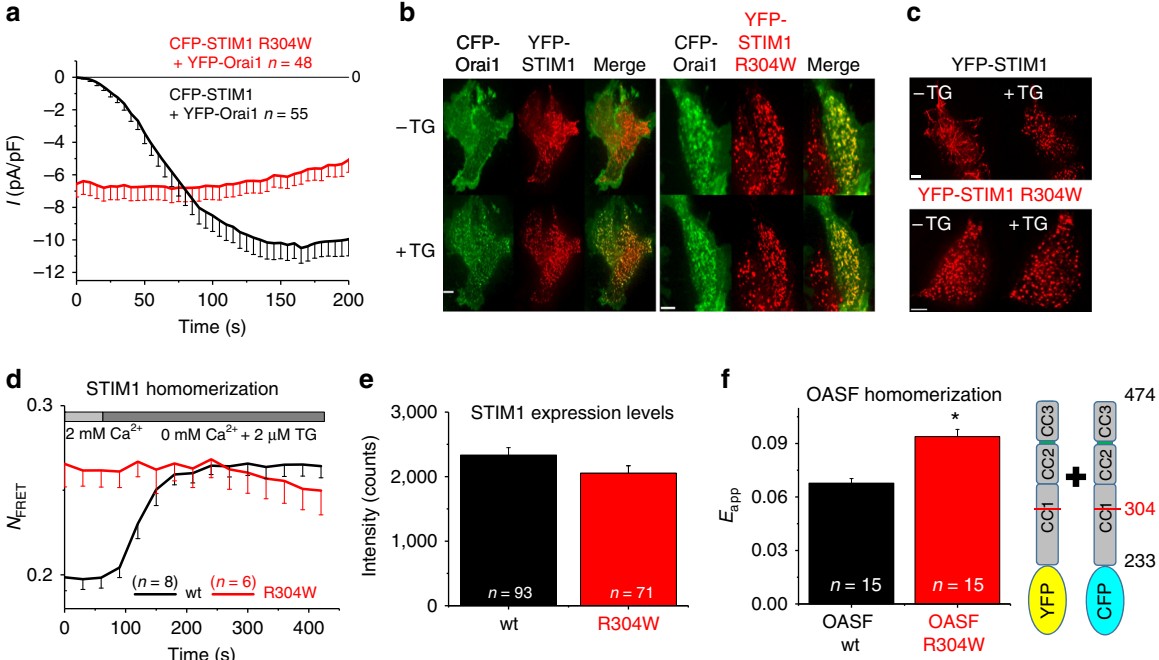

**Fig. 1** The Stormorken R304W STIM1 mutant is constitutively active. All experiments performed with HEK 293 cells. **a** Patch clamp recordings of CFP-STIM1 + YFP-Orai1 (black) and Stormorken mutant CFP-STIM1 R304W + YFP-Orai1 (red). **b** Representative images of co-localization experiments using confocal fluorescence imaging of CFP-Orai1 + YFP-STIM1 (left) and CFP-Orai1 + YFP-STIM1 R304W (right), both in resting (−TG) as well as store depleted (+TG) state. Length of scale bars correspond to 5 μm. **c** Representative images of localization experiments using confocal fluorescence imaging of YFP-STIM1 (top) and YFP-STIM1 R304W (bottom), both in resting (−TG) as well as store depleted (+TG) state. Length of scale bars correspond to 5 μm. **d** FRET homomerization experiments of YFP- + CFP- labeled STIM1 (black) and YFP- + CFP-labeled STIM1 R304W (red) in response to stored depletion using 2 μM TG. **e** Expression levels of STIM1 wt and STIM1 R304W, respectively, in HEK 293 cells. **f** FRET homomerization experiments of YFP-OASF + CFP-OASF (black) and YFP-OASF R304W + CFP-OASF R304W (red), respectively. *Significant difference ($p < 0.05$). Error bars are defined as SEM. Statistics are Student's $t$-test

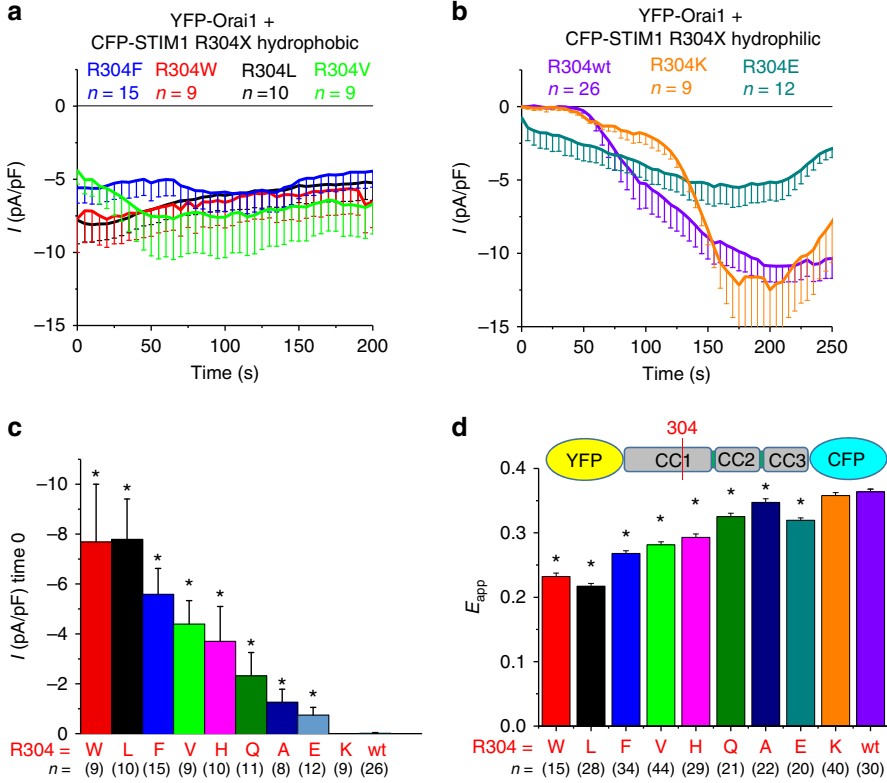

**Fig. 2** STIM1 R304W impact of hydrophobicity. All experiments performed with HEK 293 cells. **a** Patch clamp recordings of YFP-Orai1 + CFP-STIM1 R304X hydrophobic mutants (R304X = F (blue), W (red), L (black), V (green)). **b** Patch clamp recordings of YFP-Orai1 + CFP-STIM1 R304X hydrophilic mutants (R304X = K (orange), E (cyan), wt (violet)). **c** Bar diagram of patch clamp recordings showing initial current densities of HEK 293 cells expressing YFP-Orai1 + CFP-STIM1 R304X (R304X = W (red), L (black), F (blue), V (green), H (magenta), Q (olive), A (marine blue), E (cyan) K (orange), wt (violet)). *Significant difference ($p < 0.05$) to wt. **d** FRET measurements of HEK 293 cells expressing conformational sensor YFP-OASF-CFP R304X mutants. The color code corresponds to **c**. *Significant difference ($p < 0.05$) to wt. Error bars are defined as SEM. Statistics are Student's $t$-test

Our data suggest that a localized helical extension of the CC1α2–CC1α3 linker within the CC1 domain together with increased homomerization propensity induced by the R304W mutation switches STIM1 into the extended, activated conformation. This dual mechanism is distinct from other mutations in the initial portion of the CC1 domain that directly weaken the CC1–CC3 clamp when inducing STIM1 into a similar extended, active state. In aggregate, the rigidity of CC1α2–CC1α3 linker of wild-type STIM1 is fine-tuned for physiological signaling by enabling a STIM1 homomerization propensity and SOAR/CAD exposure precisely situated between constitutive and significantly slowed store-dependent activation.

## Results

### The R304W STIM1 mutant adopts a constitutively active state.
In an attempt to initially characterize the function of the Stormorken-related STIM1 R304W, we co-overexpressed YFP-Orai1 and CFP-STIM1 R304W in HEK293 cells for electrophysiological recordings. Results revealed constitutive inward $Ca^{2+}$ currents (Fig. 1a), suggesting that the Stormorken R304W mutation induced an active conformation of STIM1, which constitutively activated Orai1 in the PM in contrast to the store-dependent behavior of wild-type STIM1. Indeed, confocal fluorescence imaging of YFP-STIM1 R304W and CFP-Orai1 co-expressing cells yielded co-localization of these proteins before and following thapsigargin (TG) application, pointing to a store-independent STIM1 R304W–Orai1 coupling (Fig. 1b). Moreover, even without Orai1 co-expression, the YFP-STIM1 R304W mutant exhibited a clustered distribution independent of store depletion, in contrast to the typical puncta formation in YFP-

STIM1-expressing cells following TG application (Fig. 1c). Correspondingly, FRET analysis of co-expressed wild-type CFP-STIM1 and YFP-STIM1 clearly showed the STIM1 homomerization-related FRET increase following store depletion (Fig. 1d). In contrast, FRET from STIM1 R304W-expressing cells showed at comparable expression levels (Fig. 1e) already a similarly high value that did not further increase upon store depletion (Fig. 1d) consistent with constitutive, enhanced homomerization. Further homomerization FRET analysis of co-expressed, cytosolic CFP-/YFP-OASF (STIM1 aa233–474) from wild-type or R304W mutant revealed an increased FRET with the OASF R304W protein, suggesting a higher propensity of homomerization potentially linked to the activated state of the protein (Fig. 1f). In summary, our data showed the point mutation R304W located near the C-terminal end of the CC1α2 domain promoted STIM1 homomerization, puncta formation, as well as constitutive coupling to and activation of Orai1 channels, independent of the $Ca^{2+}$ content of the ER stores.

### STIM1 R304W impact of hydrophobicity.
STIM1 R304W represents a dramatic substitution with regard to the side chain physicochemical nature. At physiological pH, wild-type R304 is positively charged and highly hydrophilic, whereas the Stormorken syndrome-related R304W mutant contains a rather bulky side chain with an aromatic double ring featuring high hydrophobicity. However, it is noteworthy that W contains one nitrogen in the first ring, which can act as hydrogen donor for hydrogen bond formation. To probe the relationship between side-chain hydrophobicity at the 304 position and STIM1 GoF, R304 was substituted by different residues with respect to

hydrophobicity. Impressively, electrophysiological measurements analyzing HEK293 cells overexpressing YFP-Orai1 and CFP-STIM1 R304X mutants (R304X = R304L, R304F, R304W, and R304V) revealed constitutive, inward-rectifying $Ca^{2+}$ currents in case of these highly hydrophobic substitutions (Fig. 2a). Of these hydrophobic substitutions, STIM1 R304V was the least stimulatory, possibly due to the smaller side chain size. In contrast, residues with distinct hydrophilic side chains lost their constitutive activity completely or almost completely (Fig. 2b). STIM1 R304K, a substitution that retained similar hydrophilicity as R304 and the permanent positive charge, was clearly store dependent; however, the activation of Orai1 current was slightly delayed in time (Fig. 2b), suggesting that STIM1 R304K was more stabilized in its quiescent state. STIM1 R304E, which represents a charge swap, lead to mild constitutive currents and decreased maximal current densities, suggesting that the negative charge at aa position 304 could impact the conformation of activated STIM1; nevertheless, R304E partially retained the ability for ER store-dependent activation. Other STIM1 R304X substitutions with uncharged side chains and less hydrophobicity (R304X = R304Q, R304A, and R304H) typically showed reduced constitutive currents (Fig. 2c and Table 1).

As the STIM1 activation status is reflected in the conformation of STIM1 C terminus[31,34–37], we analyzed the STIM1 C-terminal conformational rearrangements induced by R304X mutations, employing our double-labeled conformational sensor construct YFP-OASF-CFP[36]. The extent of intramolecular FRET between the two fluorophores was used as a reporter of structural transitions. Wild-type OASF representing the tight, quiescent STIM1 conformation yielded the expected high FRET. In contrast, the activated STIM1 cytosolic part is represented by an extended structure as previously shown by introduction of the activating mutation L251S[34,36,37]. The Stormorken-derived conformational sensor YFP-OASF-CFP R304W yielded a low FRET similar to YFP-OASF-CFP L251S, suggesting an extended conformation of the STIM1 C-terminal portion consistent with its constitutive activity. In line with patch clamp recordings, which showed constitutive activation of CRAC channels, hydrophobic substitutions (R304X = R304W, R304L, R304F, and R304V) in YFP-OASF-CFP (Fig. 2d) exhibited an extended state. Shifting the side-chain characteristic toward hydrophilicity resulted in higher FRET values of YFP-OASF-CFP R304X (R304X = R304H, R304Q, R304A, and R304K) (Fig. 2d), suggesting a switch to the tighter, inactive STIM1 conformation (Table 1).

In summary, the results from Fig. 2c and Fig. 2d indicated that the physicochemical nature of the side chain at position 304 was

involved in mediating the graded differences in observed constitutive current size and FRET, respectively, where hydrophobicity is a key determinant, but other factors such as side-chain size had an additional role.

**STIM1 R304W acts in a position-specific manner**. In order to probe the role of aa position 304 in STIM1 activation, we engineered different single point mutants in the range of aa 300–aa 308. One helical turn in an α-helix corresponds to 3.6 amino acids, suggesting that the side chains of aa 300/308 are on the same helical face as R304. Furthermore, we also investigated whether the direct neighbors of R304 (L303 and E305) would induce the same effect as R304W, if they harbored a W (L303W and E305W, respectively). Introducing single point mutations into the conformational sensor YFP-OASF-CFP (L300W, L303W, E305W, and E308W, respectively) revealed the high impact of R304W, a moderate effect of the direct neighbors L303W and E305W, and a very small effect with L300W and E308W (Fig. 3a). These data of the YFP-OASF-CFP mutants, obtained without Orai1 co-expression, suggested a high position specificity of R304W. The impact of Orai1 co-expression with the full-length STIM1 single point mutants (L300W, L303W, E305W, and E308W, respectively) in HEK293 cells revealed a congruent behavior in patch clamp experiments, where STIM1 R304W induced the largest constitutive current, followed by STIM1 L303W and STIM1 E305W. Store-dependent currents were recorded with STIM1 L300W and STIM1 E308W with the former showing a slower rate of activation (Fig. 3b). Confocal microscopy images depicted the clustered distribution of CFP-STIM1 R304W mutant constitutively co-localizing with YFP-Orai1, with no obvious changes following store depletion, whereas CFP-STIM1 R303W reacted to TG moving from partial co-localization to STIM1 wild-type-equivalent co-localization with YFP-Orai1 in the store-depleted state (Fig. 3c, d). Detailed analysis of co-localization of STIM1 wild type or mutants with Orai1 before and after ER store depletion with TG (Fig. 3d) indicated for STIM1 R304W no dependence on store depletion, which matched both the constitutive Orai1 channel activation (Fig. 3b) and the extended conformation of YFP-OASF-CFP R304W (Fig. 3a). Consistent with the patch clamp data, STIM1 L303W and E305W also formed puncta and co-localized with Orai1 in a store-independent manner, however, to a lesser extent compared with STM1 R304W. Clear store-dependent co-localization with Orai1 similar to wild-type STIM1 was found with STIM1 L300W and E308W, which coincided with the conformational sensor data and electrophysiological measurements (Fig. 3a, b, d).

In summary, experimental approaches utilizing FRET, confocal microscopy, and patch clamp revealed that the Stormorken STIM1 R304W mutation acted in a position-specific manner, as W substitutions of adjacent residues exhibited a decreasing ability to constitutively activate CRAC channels with increasing distance from R304.

**Gain of function of STIM1 R304W independent of Q314 E318**. Previously, X-ray crystallographic resolution of STIM1 CC1 fragments has revealed an antiparallel dimerization of CC1, with α2α3 interacting reciprocally[40]. Based on these structural data, residue R304 of one monomer is in close proximity with Q314 and E318 of the other monomer, forming a hydrogen bond and an ionic bond between R304 and Q314, E318, respectively. The Stormorken-related mutant STIM1 R304W has been therefore hypothesized to result in GoF due to the interrupted interactions with Q314 and E318 (Fig. 4a)[21]. To test this potential mechanism, we introduced several single point mutations (Q314W, Q314K, E318W, and E318K) in the conformational sensor YFP-OASF-

**Table 1 R304 mutants depicted according to their substitution and qualitatively assessed by their hydrophobicity, constitutive current, and YFP-OASF-CFP extension**

| R304X substitution | Hydrophobicity | Constitutive current | YFP-OASF-CFP extension |
| --- | --- | --- | --- |
| R304W | ++ | ++ | ++ |
| R304L | ++ | ++ | ++ |
| R304F | ++ | ++ | ++ |
| R304V | ++ | ++ | ++ |
| R304H | + | + | + |
| R304A | + | + | + |
| R304Q | + | + | + |
| R304E | − − | + | + |
| R304K | − − | − − | − − |
| wt R304 | − − | − − | − − |

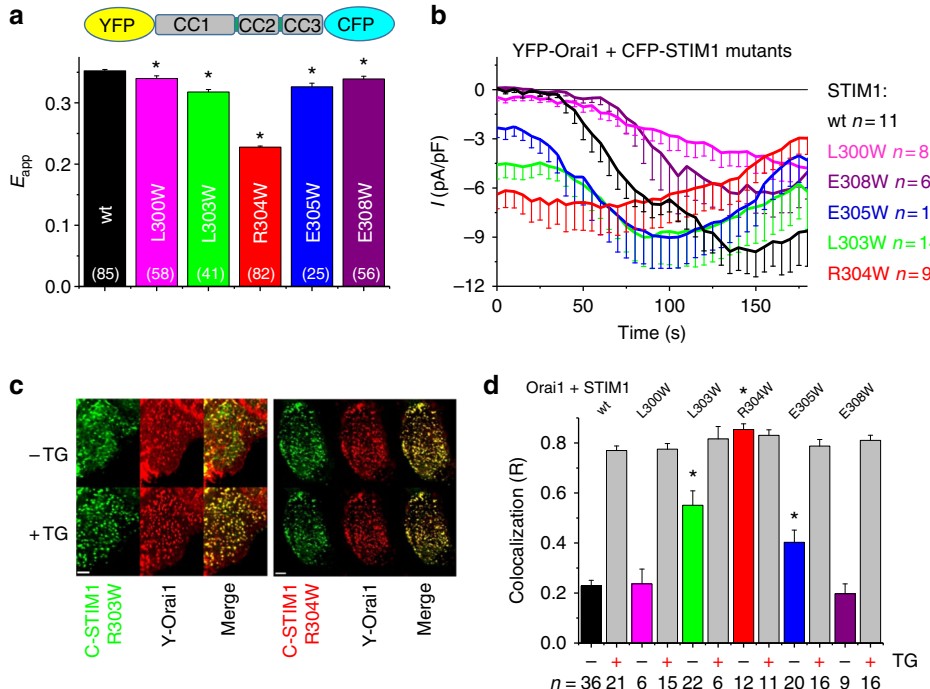

**Fig. 3** High specificity of STIM1 amino acid position 304. All experiments performed with HEK 293 cells. **a** Bar diagram of FRET measurements of conformational sensor YFP-OASF-CFP wt (black) and mutants (L300W (magenta), L303W (green), R304W (red), E305W (blue), E308W (violet)). *Significant difference ($p < 0.05$) to wt. n-number in brackets. **b** Patch clamp recordings of YFP-Orai1 + CFP-STIM1 wt (black) and mutants (color code corresponds to **a**). **c** Representative confocal images of cells co-expressing CFP-STIM1 R303W + YFP-Orai1 (left) and CFP-STIM1 R304W + YFP-Orai1 (right) in resting (−TG) as well as store-depleted (+TG) conditions. Merge shows localization of both CFP-STIM1 and YFP-Orai1 signals in one cell. Length of scale bars correspond to 5 μm. **d** Bar diagram representing co-localization experiments using confocal microscopy of HEK293 cells co-expressing YFP-Orai1 and CFP-STIM1 wt and mutants, respectively, in both resting (−TG) as well as store-depleted (+TG) conditions (color code corresponds to **a**). *Significant difference ($p < 0.05$) to wt (only −TG). Error bars are defined as SEM. Statistics are Student's t-test

CFP. If the interrupted electrostatic interactions between R304 and Q314, E318 were the cause for the extended OASF conformation, we expected the Q314W, Q314K, E318W, E318K YFP-OASF-CFP mutants to behave similar to YFP-OASF-CFP R304W. However, the Q314W, Q314K, E318W, and E318K mutants yielded FRET values in the range of wild-type OASF (Fig. 4b), suggesting minimal impact of the speculated interactions between R304 and Q314, E318 on the STIM1 activation state. In addition, we co-expressed full-length CFP-STIM1 Q314 and E318 mutants, respectively, with YFP-Orai1 in HEK293 cells for electrophysiological analyses. In line with the conformational sensor FRET data, all of the STIM1 mutants (Q314W, Q314K, E318W, and E318K) clearly revealed store-dependent activation of Orai1 in patch clamp experiments with some slight variations in the rate of activation (Fig. 4c). Furthermore, to fully avoid potential side-chain hydrogen bond formations of substituted residues, we analyzed CFP-STIM1 Q314A and E318A mutants in electrophysiological experiments, both of which resulted in clear, store-dependent Orai1 activation (Supplementary Fig. 1) similar to the experiments performed with the other STIM1 mutants (Q314W, Q314K, E318W, and E318K).

In summary, our data suggested it as unlikely that an R304W-mediated disruption of R304:Q314/E318 interactions led to constitutive STIM1/Orai1 activation; however, additional high-resolution structural information of the full cytoplasmic domain in the active and inactive conformations is required to precisely assess the significance of the Q314 and E318 residues, and their potential interactions in the STIM1 activation state.

**R304W does not perturb interaction of CC1 and CC3 fragments.** To understand what drives STIM1 R304W into the

extended conformation, we approached the mechanism underlying the GoF with our previously reported "FIRE" system[34]. "FIRE" has been developed to evaluate protein fragment interactions in a native ER-tethered environment. The FIRE constructs (see Fig. 5a) consist of STIM1 domains of interest linked via a flexible, 32 glycine linker to the ER STIM1 TM domain, which is fused to a fluorescent tag (yellow fluorescent protein (YFP) or cyan fluorescent protein (CFP)) on the ER luminal side[34]. Previously, we reported on a coiled-coil clamp between CC1 and CC3 domains in controlling the cytosolic STIM1 activation status. In order to examine the impact of the R304W mutation on the CC1-CC3 clamp interaction, which, under wild-type resting conditions, constituted the tight, quiescent state, we co-expressed FIRE constructs of CC3 and CC1 wild type and mutants. As expected, CC3 strongly interacted with CC1, mimicking the tight state of STIM1 C terminus (Fig. 5a, black bar). Employing CC1 L251S together with CC3 showed strongly decreased interaction (Fig. 5a, blue bar), in line with the previously reported extended conformation of YFP-OASF-CFP L251S and GoF of full length STIM1 L251S. Surprisingly, considering the overall equivalent behavior of YFP-OASF-CFP L251S and YFP-OASF-CFP R304W in showing decreased FRET (see Fig. 5b), FIRE experiments employing CC3 and CC1 R304W revealed a robust interaction similar to wild type, suggesting that CC1 R304W did not have a perturbed affinity for CC3 (Fig. 5a, red bar). However, OASF R304W existed in the extended state even though the CC1 R304W-CC3 interaction per se was apparently not impaired in FIRE experiments in contrast to the CC1 L251S mutation. Most likely, the CC1 R304W and CC3 fragments in FIRE had enough degrees of freedom for their interaction, which might be disturbed in the whole OASF R304W entity, possibly due to sterical

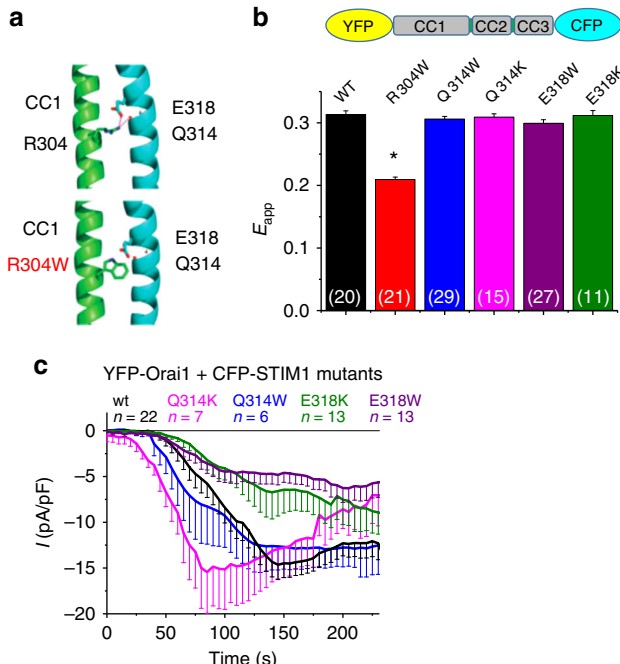

**Fig. 4** Constitutive activity of STIM1 R304W independent of Q314 E318. **a** Cartoon representing a crystallized STIM1 CC1 dimer[21, 40] with highlighted interaction between R304 or R304W, respectively, with – Q314 and −E318[21]. **b** Bar diagram of FRET measurements of HEK 293 cells expressing conformational sensor YFP-OASF-CFP wt (black) and mutants (R304W (red), Q314W (blue), Q314K (magenta), E318W (violet), E318K (green)). *Significant difference ($p < 0.05$) to wt. n-number in brackets. **c** Patch clamp recordings of HEK 293 cells co-expressing YFP-Orai1 + CFP-STIM1 wt (black) or mutants (color code corresponds to **b**). Error bars are defined as SEM. Statistics are Student's t-test

constraints that impeded formation of the optimal CC1-CC3 clamp interface. Generating a double mutant by combining both L251S and R304W in YFP-OASF-CFP resulted in even lower FRET values, pointing to additive effects of the mutations probably due to distinct molecular mechanisms (Fig. 5b).

In previous reports, we have identified the point mutation R426L located within CC3 that strengthens the heptad repeat of CC3 resulting in an enforced CC1-CC3 clamp interaction[34,36]. In full-length STIM1 as well as OASF, R426L stabilizes the quiescent, tight state, yielding substantially reduced coupling to and activation of Orai1 channels[36]. Combining R304W and R426L in YFP-OASF-CFP indeed rescued the tight state, suggesting that the double mutant, by strengthening the CC1-CC3 interaction with R426L, counteracted the R304W effect (Fig. 5b). Patch clamp experiments with HEK293 cells co-expressing YFP-Orai1 and CFP-STIM1 wild type and mutants, respectively, revealed similar store-independent, constitutive Orai1 activation in presence of STIM1 L251S or STIM1 R304W, in line with the conformational sensor data (Fig. 5c). The double L251S R304W mutation, which showed an additive effect in YFP-OASF-CFP analyses, had no additional effect in electrophysiological measurements (Fig. 5c). It should be noted that the conformational sensor data, as recorded in the absence of Orai1, might not fully reconstitute effects obtained in patch clamp experiments. In patch clamp recordings, STIM1 was likely fully extended and activated in the Orai1 coupled state[36], which surpassed that effect of L251S or R304W visible in the conformational sensor data. The combination of R304W and R426L mutation in STIM1 resulted in substantial reduction of constitutive Orai1 current activation followed by additional store-

dependent activation (Fig. 5c) consistent with a potentially functioning, yet R304W-disturbed formation of the CC1–CC3 clamp in controlling the STIM1 activation state.

In summary, both L251S and R304W mutations, respectively, resulted in GoF of STIM1, with both substitutions inducing the extended conformation of OASF and constitutive activation of Orai1. However, the mechanisms underlying the trigger of structural extension were distinctly different as shown by FIRE measurements. L251S disrupted the CC1–CC3 clamp interaction, releasing the tight state, contrary to R304W, which did not diminish the affinity of CC1 to CC3 fragments per se. Most probably, R304W, by utilizing a distinct molecular mechanism than L251S, impeded formation of the CC1–CC3 clamp within the OASF entity resulting in the extended, activated state of STIM1.

To assess the biochemical effects of R304 variations on the CC domains in isolation, we expressed and purified wild-type and R304 mutant versions of STIM1 234–491 corresponding to an extended OASF fragment (OASF_ext). We evaluated the thermal stability of the R304A and R304W OASF_ext proteins by monitoring the loss in far-UV circular dichroism (CD) signal at 222 nm as a function of temperature. The thermal melts revealed that R304A and R304W OASF_ext compared with wild-type OASF_ext exhibited a slightly lower apparent midpoint of temperature denaturation by $\sim$− 2 °C (Supplementary Fig. 2a). The destabilization observed for these R304 mutants is in-line with the destabilization previously observed for L251S, L416S, and L423S OASF_ext mutations which induce a conformational extension[36]. Thus, to qualitatively evaluate the conformation of the OASF_ext R304 mutants, we performed size exclusion chromatographic analysis. The R304W and R304A mutant versions of the STIM1 OASF_ext eluted at earlier volumes compared to wild-type (Supplementary Fig. 2b). We have previously shown that wt OASF_ext elutes as a dimer and observed that shifts to earlier volumes are the result of a conformational extension[17,36], consistent with the R304W and R304A mutants inducing maximum or partial constitutive Orai1 channel activity, respectively (Fig. 2c). In cells, the R304A mutant constitutively accessed both extended and quiescent conformation inducing only a partial activation (Fig. 2c); in isolation, the extended conformation for R304A is favored, perhaps due to the absence of macromolecular crowding effects and other differences endowed by the intracellular milieu.

**CC1 R304W increases CC1–CC1 homomerization.** Besides the regulatory impact of the CC1 interaction with CC3, we and others have reported the CC1 homomerization additionally contributes to STIM1 activation[34,37]. With respect to the Stormorken-related STIM1 R304W mutant, we next focused on the impact of the R304W mutation on CC1 homomerization as an alternative mechanism promoting the STIM1 extended, active state. Employing the FIRE system, CC1 wild type revealed strong homomerization potential as evident from FRET measurements (Fig. 6b). By comparing the effects of both L251S or R304W mutations on CC1 homomerization, respectively, the two mutations displayed opposite effects. Although CC1 L251S depicted diminished homomerization, CC1 R304W exhibited an increased homomerization potential (Fig. 6b). Obviously, both L251S and R304W mutations drive opposing forces which act on CC1 homomerization. Combination of these substitutions in CC1 L251S R304W demonstrated the antagonistic effects of the two mutations on CC1 homomerization (Fig. 6b). Nonetheless, it is important to note that the dominant effect of the L251S mutation is the release of the CC1–CC3 clamp and switching STIM1 into an extended, active state, despite the decrease of the CC1 homomerization[34].

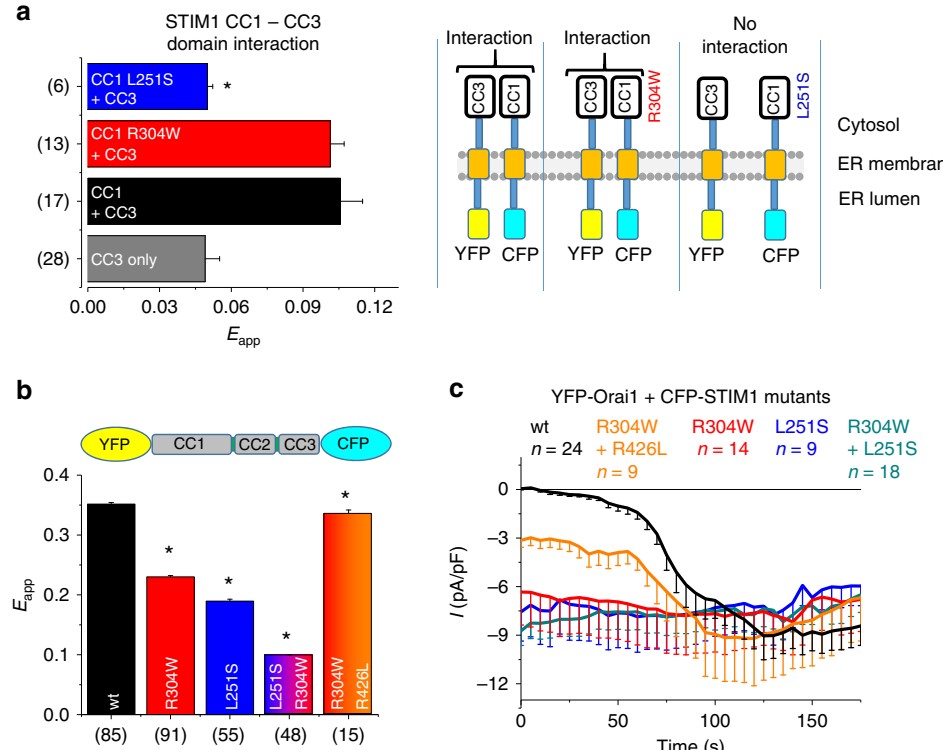

**Fig. 5** STIM1 CC1 R304W does not perturb the CC1–CC3 fragment interaction. All experiments performed with HEK 293 cells. **a** Bar diagram of FRET analysis (FIRE system) showing CTMG-CC3 + YTMG-CC1 wt (black) and mutants (red, R304W; blue, L251S), respectively. Control (gray) represents YTMG-CC1 co-expressed with empty CTMG. *Significant difference ($p < 0.05$) to (CC1 + CC3). n-number in brackets. **b** YFP-OASF-CFP conformational sensor FRET analysis with wt (black), R304W (red), L251S (blue), L251S + R304W (red + blue), and R304W + R426L (red + orange). *Significant difference ($p < 0.05$) to wt. n-number in brackets. **c** Patch clamp recordings of HEK cells overexpressing YFP-Orai1 + CFP-STIM1 wt (black) or mutants (R304W; L251S; double L251S + R304W; and double R304W + R426L). Error bars are defined as SEM. Statistics are Student's t-test

To further explore the homomerization enhancing effect of R304W, we have analyzed shorter CC1 variants (233–309 CC1α1α2 and 281–342 CC1α2α3) with the same set of point mutations. Indeed, CC1α1α2 FIRE measurements led to similar results compared to the full CC1 domain (i.e., wild type revealed strong homomerization, L251S decreased the interaction, and R304W resulted in enhancement of CC1α1α2–CC1α1α2 interplay) (Fig. 6c). In line, the double mutant CC1α1α2 L251S R304W correlated with CC1 data, demonstrating the rescuing effect of R304W on CC1α1α2–CC1α1α2 interaction (Fig. 6c). These data suggest, that R304W, which is at the very end of CC1α2, promoted CC1 homomerization. Further analyses employing α1-deficient CC1α2α3 and CC1α2α3 R304W constructs revealed no impact of R304W on interaction, strengthening the idea that CC1α1 plays a dominant role in R304W enforced homomerization (Fig. 6c). Employing the highly similar substitution R304K in CC1 and CC1α1α2, respectively, demonstrated in both cases a slightly decreased homomerization (Fig. 6b and c), compatible with the slower development of store-dependent Orai1 currents (see Fig. 2b).

Finally, to biochemically (Supplementary Methods) assess the propensity for protein–protein interactions, we subjected wild-type and R304W OASF$_{ext}$ proteins to glutaraldehyde crosslinking. Indeed, R304W OASF$_{ext}$ demonstrated significantly reduced persistence of monomer compared to wt proteins upon glutaraldehyde incubation, indicating a higher propensity for self-association (Supplementary Fig. 2c). A distinct oligomeric species could not be discerned from the SDS-polyacrylamide gel electrophoresis assessment of the crosslinking, suggesting a heterogeneous mixture of oligomeric states (Supplementary

Fig. 2d). To better evaluate differences in the affinities between wt STIM1 and STIM1 R304W, we performed analytical ultracentrifugation (AUC) experiments with OASF$_{ext}$ wt, L251S and R304W. Using AUC, we found that wt, L251S, and R304W OASF$_{ext}$ proteins were all in a primarily dimeric state at ~0.5 mg ml$^{-1}$ in solution (Supplementary Fig. 3a–c). The dimer dissociation constants ($K_d$) were all sub-μM, suggesting high-affinity monomer interactions. Estimation of the self-association constants of the dimer unit building blocks (i.e., dimer-to-tetramer association) revealed more notable differences. Although the self-association affinity of the dimers for all the proteins were sub-mM, the R304W protein showed a relatively higher self-association affinity than the wt protein (i.e., $K_d$ ~144 ± 27 versus ~265 ± 116 μM for R304W, respectively; Supplementary Fig. 3d–f). This apparently higher affinity is consistent with the less persistent monomer bands for the R304W protein compared with wt in our crosslinking experiments (Supplementary Fig. 2c, d). The L251S protein also showed an apparently higher affinity compared with wt in the OASF$_{ext}$ context. This increased affinity is in line with the promoted intermolecular CC3–CC3 interactions after release of the CC1–CC3 clamp, which leads to Orai1 activation[34]. Given the decreased CC1–CC1 interactions caused by L251S (see Fig. 6), the higher apparent affinity of L251S OASF$_{ext}$ compared with wt is likely driven through the promoted CC3–CC3 interactions. We have previously shown that CC3 has the highest propensity for homomeric STIM1 coiled-coil interactions using our FIRE technique[34].

In summary, R304W in STIM1 significantly enhanced CC1 homomerization, which contributed to the destabilization of the CC1–CC3 clamp probably caused by an unfavorable CC1–CC3

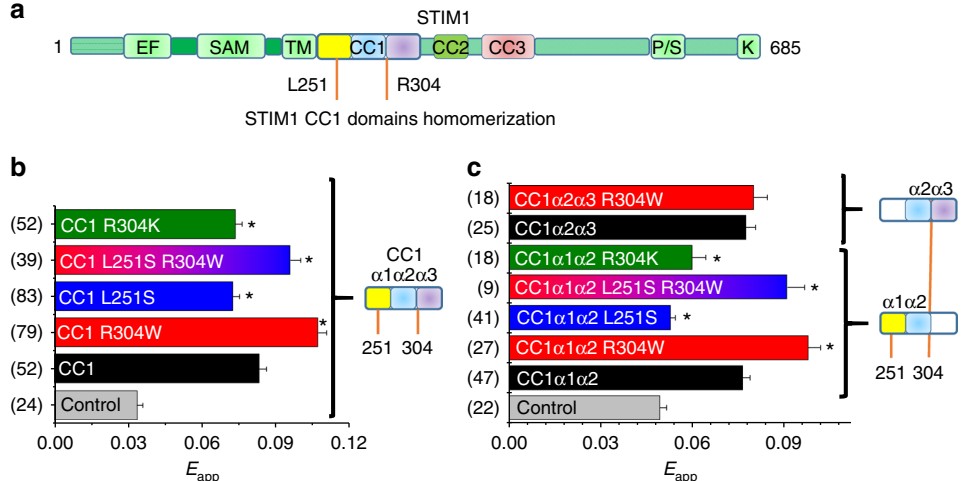

**Fig. 6** CC1 R304W increases CC1-CC1 homomerization. All experiments performed with HEK 293 cells. **a** Schematic representation of STIM1. **b** Bar diagram of FRET analysis (FIRE system) showing homomerization of CC1 wt or mutants (R304W, L251S, double L251S + R304W, R304K). Control (gray) represents YTMG-CC1 co-expressed with empty CTMG. *Significant difference (p < 0.05) to CC1 (black). **c** FRET analysis (FIRE system) of homomerization experiments of CC1α1α2 wt or mutants (R304W, L251S, double L251S + R304W, R304K) (*significant difference (p < 0.05) to CC1α1α2 (black)); CC1α2α3 wt and R304W, respectively. n-number in brackets. Error bars are defined as SEM. Statistics are Student's t-test

binding interface in full-length STIM1 (and OASF). The disturbance of the intramolecular CC1–CC3 clamp formation finally yielded the extended, activated STIM1 state.

**The R304W mutation promotes extension of the CC1α2 helix.** To analyze in more detail the local effect of the R304W mutation, we additionally performed NMR experiments with purified recombinant [15]N-labeled CC1 fragments and mutants. The results of [15]N longitudinal relaxation experiments indicated that the rigidity of the E305–E310 stretch following the R304W site in CC1 increased significantly and the longitudinal relaxation time at residue E310 doubled compared with CC1 wt (Fig. 7a). This increase of amide-[15]N-$T_1$ toward values typical of α-helices can be interpreted as an elongation of the α2 helix by ~1.5 additional turns. Bioinformatics secondary structure predictions (Fig. 7b) of CC1 wt supported this result, suggesting a high random coil propensity for aa 306–313, the linker region between CC1α2 and CC1α3. By contrast, in CC1 R304W the α2 helix was extended in the prediction, thus reducing the random coil portion 305–311, consistent with the NMR data. Mutating CC1 residues 305–311 to Ala (i.e., (305–311)A) was predicted to increase α-helical structure formation in the linker between CC1α2 and CC1α3, whereas the combination of R304W + (308–310)G mutations was likely to interfere with the helix extension as seen in CC1 R304W (Fig. 7b) [41]. Based on these data, we engineered a multiple alanine mutant (305–311)A in CFP-STIM1 as well as in YFP-OASF-CFP, to promote ~1.5 additional α-helical turns in CC1α2. Electrophysiological measurements of CFP-STIM1 (305–311)A and YFP-Orai1 co-expressing HEK cells yielded constitutive currents similar to STIM1 R304W (Fig. 7c and d). The equivalent effect of STIM1 R304W and STIM1 (305–311)A was consistent with the NMR data reinforcing the notion that R304W induces an increase of α-helical propensity. In line with these results, a CFP-STIM1 R304W (308–310)G mutant, which should have significantly less α-helix character [41], resulted in a decrease of constitutive CRAC channel activation (Fig. 7c and d). Consistently, the conformational sensor YFP-OASF-CFP (305–311)A exhibited low FRET similar to YFP-OASF-CFP R304W, indicative of the activated extended conformation. However, the YFP-OASF-CFP R304W (308–310)G resulted in higher FRET in line with the decreased constitutive CRAC channel activation (Fig. 7e). We also probed

the function of STIM1 (308–310)G as well as OASF (308–310)G in the absence of the R304W mutation, providing higher flexibility to the CC1α2–CC1α3 linker. Electrophysiology of HEK cells co-expressing CFP-STIM1 (308–310)G and YFP-Orai1 revealed clear, store-dependent $Ca^{2+}$ currents that activated at a significantly slower rate and less efficiently as wild-type STIM1 (Fig. 7c, d). YFP-OASF-CFP (308–310)G resulted in high FRET compatible with a tight OASF state (Fig. 7e). FRET-based homomerization experiments with full length YFP-/CFP-STIM1 (308–310)G revealed a decreased homomerization propensity (Supplementary Fig. 4), suggesting that the higher flexibility in the CC1α2–CC1α3 linker region weakens STIM1 homomerization compatible with the delayed and reduced Orai1 current activation (Fig. 7c)

In addition, we assessed α-helicity of wt, R304A, and R304W OASF$_{ext}$ using far-UV CD spectroscopy. The far-UV CD spectrum of the R304W protein exhibited a trend toward enhanced negative ellipticity at 208 and 222 nm, suggesting increased levels of α-helix compared with the wt protein (Supplementary Fig. 2e). This trend was in contrast to the R304A version of OASF$_{ext}$, which showed similar α-helical levels to wt (Supplementary Fig. 2e). These observations correlated with our patch clamp experiments using full-length CFP-STIM1 mutants, where R304W produced ~8-fold higher constitutive currents compared with the low levels of R304A (Fig. 2c).

In summary, the R304W mutation, besides promoting CC1 homomerization, induced a stiffening effect on the amino acid stretch between CC1α2 and CC1α3, which is likely due to a helical extension; this structural and dynamical change provided sterical constraints that altogether destabilize the CC1–CC3 clamp formation, resulting in CAD exposure within OASF, as well as full-length STIM1 and constitutive CRAC channel activation.

## Discussion

Diseases belonging to the CRAC channelopathy family can be based on both loss of function (LoF) as well as GoF mutations of CRAC components, respectively [18]. Patients with LoF mutations suffer of symptoms comprising immune dysregulation, defects in muscles, as well as ectodermally derived tissues. On the other hand, GoF mutations leading to channelopathy are represented

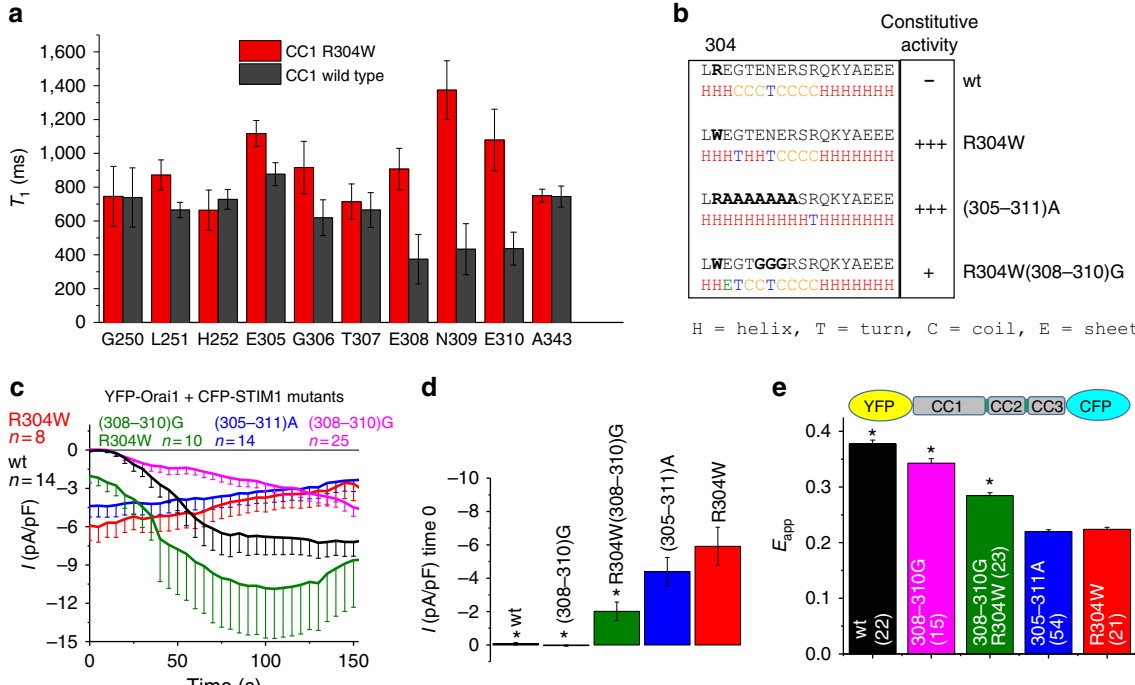

**Fig. 7** R304W induces formation of additional helical turn. **a** NMR [15]N-T1 relaxation times (70 MHz) for peptide bond [15]N of selected assigned residues in CC1 wt (black bars) and CC1 R304W (red bars). Error bars are defined as SD. **b** Bioinformatic secondary structure prediction highlighting CC1 region aa303–320. Predictions performed with sequences of wt, R304W, (305–311)A, and R304W (308–310)G, respectively. **c** Patch clamp recordings of HEK 293 cells co-expressing YFP-Orai1 + CFP-STIM1 wt (black) and mutants (R304W (red); R304W (308–310)G (green); (305–311)A (blue); (308–310)G (magenta)). Color-coded bar diagram of initial currents at time 0 of patch clamp recordings are represented in **d**. *Significant difference ($p < 0.05$) to R304W. **e** YFP-OASF-CFP conformational sensor FRET analysis with wt (black), (308–310)G (magenta), R304W (308–310)G (green); (305–311)A (blue), and R304W (red)). *Significant difference ($p < 0.05$) to R304W. n-number in brackets. Error bars are defined as SEM. Statistics are Student's t-test

by the Stormorken syndrome, the York platelet syndrome, and tubular aggregate myopathy. Overlapping symptoms of both GoF and LoF mutants comprise muscular defects and platelet dysfunction[18]. According to Lacruz and Feske[18], five different LoF STIM1 mutants have been described: one with a splice site mutation, two which have amino acid substitutions in the ER luminal part, and two mutants with substitutions (R429C and R426C) in the cytosolic portion of STIM1. STIM1 R429C has been extensively characterized by Maus et al.[17]. This mutant alters the CC3 structure and consequently the conformation of the STIM1 C terminus resulting in constitutive localization at ER–PM junctions. However, STIM1 R429C neither oligomerizes nor interacts with Orai1, therefore failing to activate the CRAC channel[17].

There are 12 different GoF STIM1 mutants identified to date, with only one mutant harboring a substitution in the STIM1 C-terminal part (STIM1 R304W)[18]. The Stormorken syndrome has been described in 1985; however, its correlation to STIM1 R304W is only reported very recently. Three research groups independently describe the autosomal dominant inherited defective allele in 21 patients from unrelated families[18,20–22]. In this context, STIM1 R304W acts as GoF mutant with constitutive cluster formation and Orai1 activation[20,21]. However, a mechanism by which STIM1 R304W results in a GoF is only speculated based on a crystal structure of STIM1 CC1[40]. In this structure, CC1 dimerizes, integrating intermolecular electrostatic interactions between R304 on one monomer and Q314/E318 on an adjacent monomer. The R304W substitution associated with Stormorken is hypothesized to disrupt this interaction leading to STIM1 activation[21]. However, our data here (Fig. 4) suggest that perturbation of the R304–Q314/E318 interaction is not likely to contribute to the R304W GoF mechanism.

Zhou et al.[37] have shown that bringing STIM1 CC1 N termini together by artificial crosslinking, triggered a conformational change within the STIM1 C-terminal portion. This physical apposition of N-terminally cross-linked STIM1 C terminus results in increased and stabilized CC1–CC1 interactions, and in addition, release of the SOAR/CAD domain from the clamp interaction with CC1[37]. In the present study, our data revealed enforced CC1 homomerization, induced by the R304W mutation, led to pathophysiological STIM1 activation. Thus, we suggest the Stormorken mutant STIM1 R304W activation state and conformational extension is in part caused by pathophysiological enhancement of the CC1–CC1 interaction, thereby supporting the exposure of SOAR/CAD and resulting in STIM1 activation.

Mechanistically, we also showed here that the Stormorken-related STIM1 R304W mutation increased the propensity for α-helicity in the CC1α2–CC1α3 linker region (Fig. 7). This resulted in higher rigidity of the respective CC1 portion triggering a tension induced conformational change between CC1α2 and CC1α3. These conformational constraints likely supported SOAR/CAD exposure both by impeding CC1–CC3 clamp formation and promoting CC1 homomerization (Figs. 5b and 6). Furthermore, our FIRE data using CC1 and CC3 domain fragments (Fig. 5a) revealed no interference of CC1–CC3 interaction in the presence of R304W, clearly showing that R304W endowed its effect only in context of a larger structure such as OASF or full-length STIM1.

Our work revealed that the R304W effect was related to the high hydrophobicity conferred by the tryptophan side chain. On the other hand, analysis of a highly similar substitution (R304K) in full-length STIM1 revealed delayed store-dependent Orai1 activation in patch clamp recordings (Fig. 2b), similar to our previously published data with the STIM1 deletion mutant

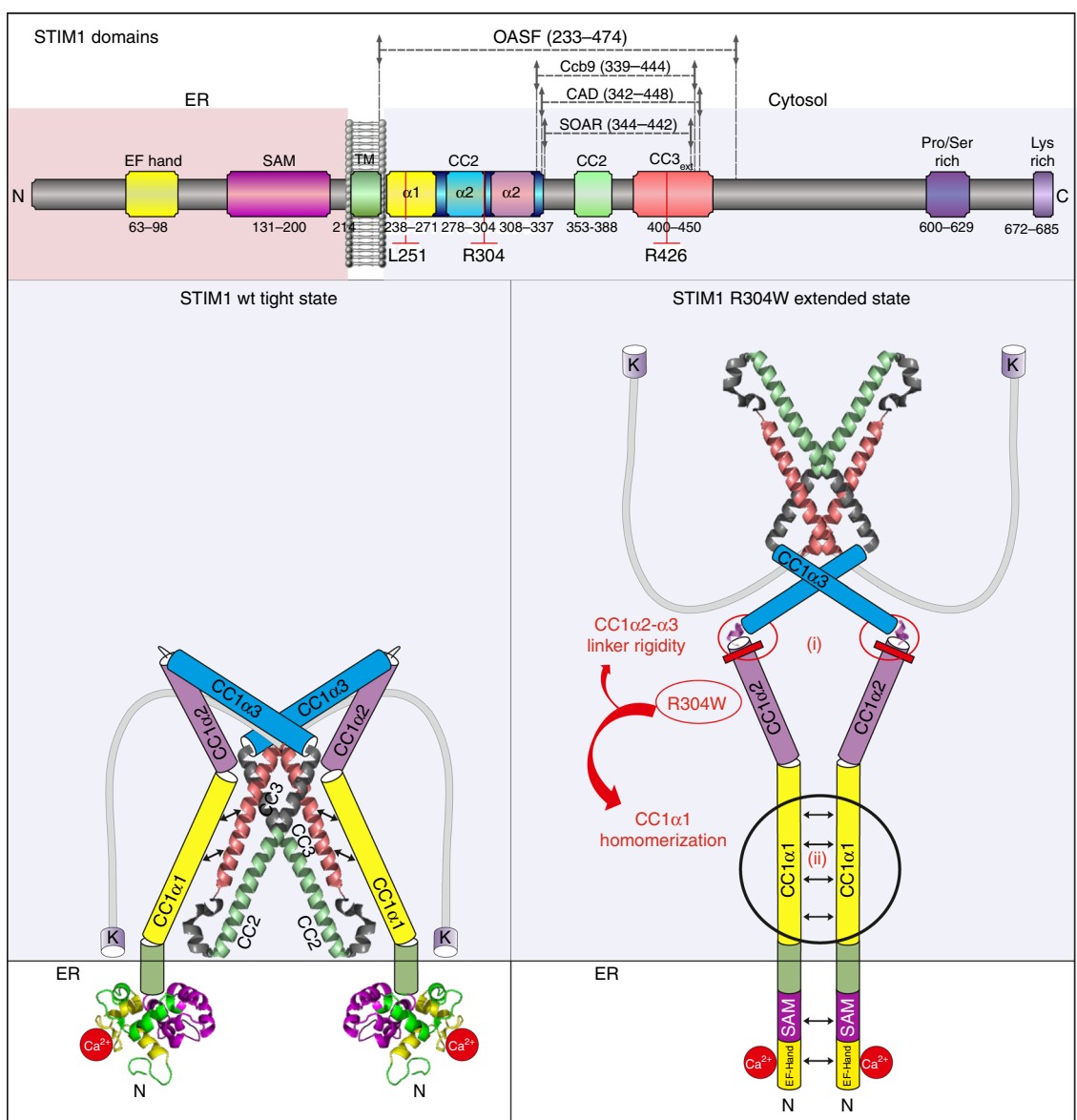

**Fig. 8** Hypothetical conformational models of STIM1 wt and STIM1 R304W in resting cell conditions. STIM1 domains (top). STIM1 wt (left) packs into a tight conformation as a result of the intramolecular CC1–CC3 clamp interaction. STIM1 R304W (right), in contrast, assumes an extended conformation. The STIM1 R304W mutation (i) increases CC1α2 helicity and stiffens the linker region between CC1α2 and CC1α3, which together with (ii) an enforced CC1–CC1 homomerization antagonizes formation of the CC1–CC3 clamp

lacking CC1α2[34]. In addition, we found that CC1 homomerization was reduced in the presence of R304K (Fig. 6b), which was in line with our electrophysiological data on STIM1 R304K as well as hypothesis by Zhou et al.[37] that CC1 homomeric interaction is an essential step preceding SOAR/CAD release. With respect to the Stormorken GoF mutant STIM1 R304W, its constitutive activity is based on two key mechanistic features (Fig. 8): (i) the helicity (and associated rigidity) of the linker region between CC1α2 and CC1α3 is increased, and (ii) the CC1–CC1 homomerization is enhanced. Both processes antagonize the CC1–CC3 clamp stability, leading to exposure of SOAR/CAD and constitutive coupling to as well as activation of CRAC channels.

An important role in the general STIM1 activation mechanism, as identified here, is obviously had by the linker region (aa304–311) connecting CC1α2 and CC1α3. This linker represents a key structural feature in modulating the switch between the quiescent, tight, and activated, extended state of STIM1. In the context of the STIM1 R304W mutant, this short stretch

gained rigidity causing constitutive exposure of the CAD/SOAR domain. Enforcing rigidity in this linker domain by introducing multiple alanines (305–311)A in STIM1 resulted in similar conformational extension and activation of STIM1. However, increasing flexibility in the linker region between CC1α2 and CC1α3 by triple glycine substitutions (308–310)G decreased STIM1 activation capabilities resulting in a reduced STIM1 (308–310)G homomerization propensity and a significantly slower rate of as well as less efficient Orai1 currents stimulation. We cannot exclude that the 308–310 triple Gly mutation and the 305–311 Ala mutations may induce additional effects beyond modulation of helix integrity, which may not occur with the R304W mutation. Nonetheless, we have identified the linker between CC1α2 and CC1α3 as an essential modulatory domain for STIM1 activation with increased and decreased rigidity enforcing and reducing STIM1 activation, respectively. Thus, the rigidity of this linker is fine-tuned for physiological signaling by enabling a wild-type STIM1 homomerization propensity and

SOAR/CAD exposure precisely situated between constitutive and significantly slowed store-dependent activation.

## Methods

**Molecular cloning and mutagenesis.** Human Orai1 (accession number NM_032790) was generously supplied by A. Rao (Harvard Medical School). N-terminally tagged Orai1 constructs were engineered by using SalI and SmaI restriction sites of the vectors pECFP-C1 and pEYFP-C1 (Clontech). N-terminally ECFP- and EYFP-labeled human STIM1 (accession number NM_003156) were made available by T Meyer (Stanford University). pECFP-STIM1 mutants (R304W, R304L, R304F, R304V, R304H, R304Q, R304A, R304E, R304K, L300W, L303W, E305W, E308W, Q314K, Q314W, Q314A, E318K, E318W, E318A, L251S, L251S_R304W, R304W_R426L, 305–311A, 308–310G, and R304W_308–310G) were built with the aid of the QuikChange XL site-directed mutagenesis kit (Stratagene). For the generation of double-tagged STIM1-OASF constructs, CFP was introduced into pEYFP-C2 via SacII and Xba1 and the OASF fragment (aa233–474) of STIM1 was inserted via restriction sites EcoRI and SacII. Double-tagged YFP-OASF-CFP mutants (R304W, R304L, R304F, R304V, R304H, R304Q, R304A, R304E, R304K, L300W, L303W, E305W, E308W, Q314K, Q314W, E318K, E318W, L251S, L251S_R304W, R304W_R426L, 305–311A, 308–310G, and R304W_308-310G) were built with the aid of the QuikChange XL site-directed mutagenesis kit (Stratagene). Constructs used for the FIRE system (Y-TMG and C-TMG) contained the STIM1-signal peptide, followed by EYFP (Y) or ECFP (C), a 29 aa linker, the STIM1 TM domain, and a linker consisting of 32 glycines followed by the protein fragment of interest (CC1 233–343; CC1_L251S; CC1_R304W; CC1_L251S_R304W; CC1_R304K; CC3 388–430; CC1α1α2_233-309; CC1α2α3_281-342). The QuikChange XL site-directed mutagenesis kit (Stratagene) was employed to generate Y- and C-TMG point mutants. For NMR experiments, STIM1 CC1 233–343 was cloned into pGEX4T1 vector. The correctness of all constructs was verified by sequence analysis. The primer sequences are listed in Supplementary Table 1.

**Confocal microscopy.** HEK-293 cells (DSMZ, Braunschweig, Germany) were analyzed by confocal FRET microscopy[42]. Specifically, a QLC100 Real-Time Confocal System (VisiTech Int.) with two Photometrics CoolSNAPHQ monochrome cameras (Roper Scientific) and a dual port adapter (dichroic: 505lp; cyan emission filter: 485/30; yellow emission filter: 535/50; Chroma Technology Corp.) was employed for obtaining fluorescence images. The system was connected to an Axiovert 200 M microscope (Zeiss, Germany) equipped with two diode lasers (445 nm, 515 nm) (Visitron Systems). The Visiview 2.1.1 software (Visitron Systems) was used for image recording and control of the confocal system. Image correction required due to cross-talk and cross-excitation was applied before FRET calculation. For this, on every day of the FRET experiment, cross-talk calibration factors were measured for each construct expressed. After threshold determination and subtraction of the background on a pixel-to-pixel basis with a custom-made software[43] integrated into MatLab 7.0.4 and based on the method reported in[44], with a microscope specific constant G value of 2.0. The experiments were carried out at room temperature. N-FRET was calculated according to Muik et al.[36] using the calibration factors a = 0.68, b = c = 0 and d = 0.36. The Pearson correlation coefficient (R-value) was used to measure the strength of the linear association/co-localization between STIM1 and Orai1 variants, where a value R = 1 means a perfect positive correlation/co-localization.

**Patch-Clamp Whole Cell Experiments.** HEK-293 cells (DSMZ) were transfected (Transfectin, Bio-Rad) with 1 µg of DNA of YFP-Orai1 and CFP-STIM1 constructs. Electrophysiological measurements were carried out following 24–34 h after transfection, employing the whole-cell recording configuration of the patch clamp technique at 21–25 °C. An Ag/AgCl electrode served as reference electrode. Every 5 s, voltage ramps covering a range of −90 to +90 mV over 1 s, were applied from a holding potential of 0 mV. For passive store depletion, the intracellular pipette solution consisted of (in mM): 145 Cs methane sulfonate, 20 EGTA, 10 HEPES, 8 NaCl, 5 MgCl$_2$ pH 7.2. The standard extracellular solution contained (in mM) 145 NaCl, 10 HEPES, 10 CaCl$_2$, 10 glucose, 5 CsCl, 1 MgCl$_2$ pH 7.4. A correction of the liquid junction potential (−12 mV) that arose from a Cl⁻-based bath solution and a sulfonate-based pipette solution, was not applied. All currents were leak corrected, by subtracting either the initial voltage ramps recorded immediately after break-in and with no visible current activation, or with constitutively active currents by subtraction of current traces obtained following 10 µM La$^{3+}$ perfusion at the end of the experiment.

**NMR methods.** All NMR experiments were carried out on 700 MHz Bruker Avance III spectrometer equipped with TCI cryogenically cooled probe at 298 K sample temperature. The sequence-specific resonance assignments were carried out by combining standard 2D $^1$H–$^{15}$N–HSQC with standard three-dimensional experiments: CbCaNH, CbCa(Co)NH,HN(Ca)CO, HNCO, HNCa, (H)C(CCO) NH, and TOCSY-HSQC. $^{15}$N longitudinal relaxation time rates were determined by exponential fitting of peak heights in series of $^{15}$N-HSQC phase-sensitive experiments (Bruker pulse program "hsqct1etf3gpsi" of TopSpin software 3.2 and

3.5pl5) with different T1 relaxation delays (10, 50, 100, 200, 300, 400, 500, 600, 700, 800, 900, 1,000, 1,100, 1,200, and 1,300 ms).

**Statistics.** The results are depicted as means ± SEM determined for the indicated number of experiments. For statistical comparison, the Student's two-tailed $t$-test was applied, considering differences statistically significant at $p < 0.05$.

**Bioinformatic secondary structure prediction.** http://www.biogem.org/tool/chou-fasman/index.php

**Data availability.** Data supporting the findings of this manuscript are available from the corresponding authors upon reasonable request. Backbone resonance assignment and longitudinal relaxation rates were submitted to BMRB database (entry 27630 and 27361 for the wild-type STIM1 and the R304W mutant, respectively).

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

## Acknowledgements

We thank S. Buchegger for excellent technical assistance. We thank Le Zheng for her technical assistance in preparing the OASFext recombinant proteins. Michael Stadlbauer and Petr Rathner have PhD scholarships of the FWF-PhD program W 1250 "NanoCell". This work was supported by Natural Sciences and Engineering Research Council of Canada (NSERC) 05239 (to P.B.S.), Canadian Institutes of Health Research (CIHR) MOP-13552 (to M.I.), and NSERC UT393093 (to M.I.). M.I. holds the Canada Research Chair in Cancer Structural Biology. In addition, support has been provided by the Austrian Science Fund (FWF) projects P28123 to M.F., P27263 to C.R., P28498 to M.M, and BMWFW HSRSM (PromOpt2.0 to C.R.).

## Author contributions

M.F. and C.R. conceived and coordinated the study, and wrote the paper. M.S. performed and analyzed electrophysiological experiments. M.M. carried out fluorescence microscopy experiments. M.F. performed molecular biology methods. P.R. and N.M. carried out and analyzed NMR experiments. P.B.S and M.I. performed and analyzed biochemical approaches. All authors reviewed the results and approved the final version of the manuscript.

## Additional information

**Competing interests:** The authors declare no competing financial interests.

