## [Peer Review File · Nature Communications]

Reviewers' comments:

Reviewer #1 (Remarks to the Author):

NCOMMS-17-16360

Fahrner et al.

In this study Dr. Romanin and co-workers have used their extensive expertise in study of structure-function relationship of STIM1 in the context of Orai1 activation to delineate the molecular remodeling of STIM1 that is induced by the R304W mutation associated with Stormorken Syndrome. This is a gain of function mutation in STIM1 which causes debilitating effects in the patients. The exact consequences of the mutation on STIM1 structure that can account for the constitutive activation of STIM1/Orai1 has not been clarified. Here the investigators have used a variety of precise molecular and functional techniques developed in their laboratory and systematic amino acid substitutions to examine the consequence of R304W on STIM1. They report that the Stormorken mutation causes an increase in the helicity of CC1 α 2 and CC1 α 3 which enhances CC1-CC1 domain interactions between STIM1 monomers. As previously shown, CC1-CC1 interactions causes remodeling of STIM1 conformation to result in exposure of the SOAR domain and activation of Orai1. Thus, R304W mutation causes activation of STIM1, recruitment into ER-PM junctions with Orai1 and activation of the channel in the absence of store-depletion. The findings are very important and not only help to further understand the cause of the disease, but also provides insight into additional structural changes within STIM1 that can contribute to its activation. Physiologically, STIM1 is held quiescent by a intramolecular CC1-CC3 clamp which is released by changes in the luminal N-terminus when ER-[Ca²⁺] is decreased. The main trigger is the increase in intermolecular CC1-CC1 interaction which releases CC1-CC3 clamp. However, the Stormorken mutation does not involve the previously identified residues involved in the CC1-CC3 clamp. These findings suggest that alterations in the helicity of the CC1 domain can also lead to CC1-CC1 interaction and activation of STIM1.

The data are very convincing, timely, and clearly presented. I only have some minor comments:

1. The authors should go over the text carefully for grammatical errors.
2. In Figure 1A, why is the current obtained with STIM1-R304W about half the size of that with WT-STIM1 and what is the slow inactivation ? Is the same inactivation seen with WT STIM1 ?
3. Figure 1C: maybe a better image for YFP-STIM1 can be provided ?
4. YFP-OASF should be in a closed conformation; if so, how does it interact with STIM1R304W ?
5. Can OASF-R304W induce constitutive activation of Orai1 or endogenous SOCE ?
6. Figure 2d, how is the graded change in intramolecular FRET explained ?
7. I suggest that the model should be shown in the main figure section.

Reviewer #2 (Remarks to the Author):

Previous studies have shown that a missense mutation (R304W) in the endoplasmic reticulum Ca²⁺ sensor, STIM1, causes constitutive activation of STIM1 and consequently Orai1 activation and store-operated Ca²⁺ entry. The constitutive SOCE is associated with a range of symptoms including severe tubular myopathy, muscle weakness, and fatigue. The present study investigates the mechanism of how R304W turns STIM1 on. This study shows that hydrophobic substitutions at R304 (e.g., L,F) cause constitutive activation of STIM1, but polar or charged substitutions are less effective (K, E, Q). Interestingly, R304W does not appear to hinder binding of CC1 with CC3, indicating that the intermolecular clamp is released through a mechanism independent of a direct disruption of SOAR-CC1 binding. Moreover, using a previously described membrane-delimited domain-binding assay (FIRE), the study shows that R304W appears to enhance the homomerization potential of CC1 domains. These effects are notably distinct from the gain-of-function phenotype caused by a previously studied L251S mutation, which directly releases

CAD/SOAR from CC1. Because R304 is located at the boundary of CC1a2 and CC1a3, Fahrner et al speculate that the R304W mutation disrupts the folded configuration of CC1a2-a3 tertiary structure, increasing the alpha-helicity of the interface at CC1a2/CCa3, thereby releasing CC1a3 and opening up the entire CC1 domain, culminating in the homomerization of CC1 and STIM1 activation.

Overall, the study is well-done, thorough, and increases our understanding of the mechanism of the R304W GOF mutation. The distinct gain-of-function mechanisms of R304W and the previously described L251S mutation are fascinating from a molecular perspective and indicate that STIM1 activation can be induced not only from a decrease in CC1-CC3 binding, but also from conformational alterations in the other domains of the intramolecular clamp, in particular, the CC1a2/CCa3 interface. However, the limitation of this study is that it is very narrowly focused on the R304W mutation. Previous work from this group and others has already shown important roles for CC1 homomerization in activating STIM1, and the summary cartoon in Supplementary Figure 1 is similar to those previously published by the group, at least in the critical part relating to the straightening of the kink at CC1a2/CCa3 (Fahrner JBC, 2014). The specific question of how precisely how the CC1a2/CCa3 kink opens up in the mutant, and how this would happen in the wild-type protein in which the alpha helicity is unchanged is not determined. Thus, the broader implications of this study for STIM1 function are somewhat incremental. I have the following additional comments:

- Figure 2. What are the effects of other key substitutions at R304? For example, Val, which is small yet highly hydrophobic (and which, according to the authors hypothesis, should also be constitutively active)? Also, Proline. Proline would be predicted to open up the CC1a2/CCa3 interface. If the authors are correct in their hypothesis that only mutations that increase alpha-helical propensity of the CC1a2/CCa3 interface should spontaneously turn on STIM1, the Pro substitution should not turn on STIM1 because even though it may open up the CC1a2/CCa3 interface, it may do little to alter the alpha-helical propensity of this region. This would distinguish between these two possibilities.

- Supplementary figure 2 the biophysical assays here do not seem to match well with the data in Figure 2. Figure 2 shows extension of OASF in R304W and almost no extension of R304A, yet the melting temperature and SEC assays show very similar results for both mutants. Glutaldehyde cross-linking gel doesn't look very convincing – a dose response curve and a negative control that does not cross-link would help. Further, given the distinct mechanisms proposed for R304W and L251S, L251S should be included in the tests shown in Supplementary Figure 2, especially panel c.

- Figure 4. The conclusion that Q314 or E318 are not involved in homomeric CC1-CC1 interactions is based entirely on the lack of GOF effects induced by W or K substitutions at these residues. Because it is well-known that W and K amino acids can also participate in hydrogen bonds, the conclusion that a previous report (ref 43) is incorrect is an over-interpretation of the very limited dataset here. It would be more convincing to show effects of hydrophobic substitutions such as Ala to test this idea.

- Figure 7b-e, the 308-310 triple Gly mutation and especially the 305-311 septuple Ala mutations are very drastic. The effects of these mutations are interpreted in the context of helix formation, but it is hard to parse out other effects in addition to helix integrity (e.g. disruption of interactions with neighboring amino acids). These mutations may not activate STIM1 through the same mechanism as R304W.

Reviewer #3 (Remarks to the Author):

This is an interesting manuscript reporting on the role of the C1 coiled-coil domains interaction in STIM1 and how the interaction is regulated by the hydrophilic environment of the C1a1 domain. The authors used the Stormorken mutation that causes gain-of-function of STIM1 to report how the mutation activates STIM1 and when doing so, discovered an important function of C1a1 in the activation of STIM1 and Ca²⁺ influx by the Orai1 channel. The findings are a major advance in our understanding of STIM1 activation and thus activation of Ca²⁺ influx and Ca²⁺-related cell functions and pathologies. The experiments are well done and provide ample structural (NMR), biochemical (FRET and CD) and electrophysiological (current) evidence to support the main conclusion.

I have only minor comments that relate to style of writing, the use of some terms and in particular the discussion. Although the manuscript is well written and the writing is mostly straight forward, the discussion does not discuss in any length the novel finding from structure-function point of view. The results section is written very well from the perspective of the significance of the R304W mutation and the hydrophilicity of C1a1 on STIM1 unfold and channel activation, this is barely mentioned in the discussion, let alone discussing whether and how the half turn extension of this domain by changing the hydrophilic environment can occur as a result of store depletion. i.e. how store depletion is transmitted to C1a1 to cause the extension, stabilize homomerization and activate STIM1. It seems that this should be the bulk of the discussion since these are the novel findings and as far as the disease is concern GOF is GOF.

Figure 2b, 3b: The behavior of R304K is interesting. Stabilization of the quiescent state may also mean that more store depletion and dissociation from STIM1 is required for its activation. This can be tested by determining if the STIM1(D76A) or other mutation that affect STIM1 Ca²⁺ binding can facilitate activation of this mutant to the same extent as wild-type. The same question is applicable to L300W and its very slow activation. The later should drive further the main novel and important message that C1a1 controls homomerization of STIM1.

Figure 4c: the significance of the effect of E318K is important and should be discussed here when describing first the mutant rather than mentioning it later in a glance.

In page 9 and later the authors suggest that R304 both stabilize homomerization and disrupts the C1-C3 clump while that data in Figure 5 provides no evidence for clump disruption. Can the authors clarify why they think or what evidence support clump disruption.

Figure 7: The control of (308-310)G alone on the current is missing.

Some of the unclear sentences/words:

Page 5, second paragraph, line 5 should read: ...noteworthy mentioning that W has/contain...

Page 8, lines 18-290: Sentence not clear and what "enough degrees of freedom" means in this context

Page 9, line 5 from bottom: add To before STIM1

Page 10, line 9: drive is not accurate, cause is better

Page 11, line9-10: should read ...the predicted α helix was extended, thus...

Page 12, line 4: Remove suffer of; line 17: change correlation to... to association with; line 22, add upon after speculated

Page 13, line9, add the afterin

Reviewers' comments:

Reviewer #1 (Remarks to the Author):

NCOMMS-17-16360

Fahrner et al.

In this study Dr. Romanin and co-workers have used their extensive expertise in study of structure-function relationship of STIM1 in the context of Orai1 activation to delineate the molecular remodeling of STIM1 that is induced by the R304W mutation associated with Stormorken Syndrome. This is a gain of function mutation in STIM1 which causes debilitating effects in the patients. The exact consequences of the mutation on STIM1 structure that can account for the constitutive activation of STIM1/Orai1 has not been clarified. Here the investigators have used a variety of precise molecular and functional techniques developed in their laboratory and systematic amino acid substitutions to examine the consequence of R304W on STIM1. They report that the Stormorken mutation causes an increase in the helicity of CC1 2 and CC1 3 which enhances CC1-CC1 domain interactions between STIM1 monomers. As previously shown, CC1-CC1 interactions causes remodeling of STIM1 conformation to result in exposure of the SOAR domain and activation of Orai1. Thus, R304W mutation causes activation of STIM1, recruitment into ER-PM junctions with Orai1 and activation of the channel in the absence of store-depletion. The findings are very important and not only help to further understand the cause of the disease, but also provides insight into additional structural changes within STIM1 that can contribute to its activation. Physiologically, STIM1 is held quiescent by a intramolecular CC1-CC3 clamp which is released by changes in the luminal N-terminus when ER-[Ca²⁺] is decreased. The main trigger is the increase in intermolecular CC1-CC1 interaction which releases CC1-CC3 clamp. However, the Stormorken mutation does not involve the previously identified residues involved in the CC1-CC3 clamp. These findings suggest that alterations in the helicity of the CC1 domain can also lead to CC1-CC1 interaction and activation of STIM1.

The data are very convincing, timely, and clearly presented. I only have some minor comments:

We thank the reviewer for her/his very good reviewing comments.

1. The authors should go over the text carefully for grammatical errors.

The text of this manuscript has been carefully amended by a native speaker.

2. In Figure 1A, why is the current obtained with STIM1-R304W about half the size of that with WT-STIM1 and what is the slow inactivation ? Is the same inactivation seen with WT STIM1 ?

We have analyzed average expression levels of wt STIM1 and STIM1 R304W in HEK cells. No significant difference was detectable between these constructs, and we have included these results in Fig. 1e (p4, l1 from the bottom). Furthermore, electrophysiological Ca²⁺ current recordings of HEK cells overexpressing Orai1 + gain of function STIM1 mutants (L251S, D76A, R304W) reveal all a diminished current size in comparison to Orai1 + WT STIM1 (Muik et al., EMBO J 30:1678, 2011 and this manuscript). Most probably, the cells afflicted with constitutive Ca²⁺ influx compensatorily show reduced current size to cope with the Ca²⁺ overload.

The slow inactivation seen with Orai1 + STIM1-R304W (Fig. 1a) was on a longer time scale also seen with Orai1 + wt STIM1 as depicted in the Fig. 1 below. As this result is not in the main scope of our manuscript, we would prefer not to include it.

Referee 1, Fig. 1

3. Figure 1C: maybe a better image for YFP-STIM1 can be provided ?

YFP-STIM1 +/- TG in Figure 1C has been substituted by a better, more representative image, as suggested by the reviewer.

4. YFP-OASF should be in a closed conformation; if so, how does it interact with STIM1R304W ?

Unfortunately, there has been a misleading color labeling in previous Figure 1e (now Fig. 1f), actually representing FRET obtained from homomerization experiments employing YFP-OASF + CFP-OASF (both wt) in comparison to YFP-OASF R304W + CFP-OASF R304W. The colors of the text underneath the bars of the diagram in Fig. 1f have been corrected. In the text of the result section we have added "homomerization FRET" (p5, l2).

5. Can OASF-R304W induce constitutive activation of Orai1 or endogenous SOCE ?

Employing patch-clamp experiments, we have analyzed CFP-OASF-R304W + YFP-Orai1 expressing HEK cells, which expectedly exhibited constitutive activation of the channel to a similar range as seen with wild-type OASF. In absence of over-expressed Orai1, CFP-OASF-R304W resulted in no Ca^{2+} current (see below Fig. 2). Due to the fact that all electrophysiological measurements throughout the manuscript have been performed with full length STIM1 constructs, we have not included the OASF measurements in the Figures.

Referee 1, Fig. 2

6. Figure 2d, how is the graded change in intramolecular FRET explained ?

We do see OASF as dynamic molecule adopting various conformations (tight and extended) over the time of FRET measurement. Depending on the mutations introduced, the OASF molecule showed this decrease in FRET compatible with higher occupancy of the extended versus tight conformation. The more time the OASF molecule stays in the extended conformation, the lower the FRET will get. Fig. 2d reveals that side chain hydrophobicity at

position 304 is a key determinant of time in the extended conformation, where the most hydrophobic side chains (i.e. Leu, Phe, Trp) (Radzicka and Wolfenden, Biochemistry 27:1664, 1988) induce the least FRET. However, side chain size may also play a role in the graded response, as Ala is also hydrophobic (albeit less than Leu, Phe and Trp) (Radzicka and Wolfenden, 1988). Thus, the physicochemical nature of the side chain at position 304 is involved in mediating the graded differences in observed FRET, where hydrophobicity is a key determinant, but other factors such as solvent accessible surface area of the side chain play also a role (p6, l15 from the bottom).

7. I suggest that the model should be shown in the main figure section.

A refined model has now been included in the main figure section as Fig. 8.

Reviewer #2 (Remarks to the Author):

Previous studies have shown that a missense mutation (R304W) in the endoplasmic reticulum Ca²⁺ sensor, STIM1, causes constitutive activation of STIM1 and consequently Orai1 activation and store-operated Ca²⁺ entry. The constitutive SOCE is associated with a range of symptoms including severe tubular myopathy, muscle weakness, and fatigue. The present study investigates the mechanism of how R304W turns STIM1 on. This study shows that hydrophobic substitutions at R304 (e.g., L,F) cause constitutive activation of STIM1, but polar or charged substitutions are less effective (K, E, Q). Interestingly, R304W does not appear to hinder binding of CC1 with CC3, indicating that the intermolecular clamp is released through a mechanism independent of a direct disruption of SOAR-CC1 binding. Moreover, using a previously described membrane-delimited domain-binding assay (FIRE), the study shows that R304W appears to enhance the homomerization potential of CC1 domains. These effects are notably distinct from the gain-of-function phenotype caused by a previously studied L251S mutation, which directly releases CAD/SOAR from CC1. Because R304 is located at the boundary of CC1 2 and CC1 3, Fahrner et al speculate that the R304W mutation disrupts the folded configuration of CC1 2- 3 tertiary structure, increasing the alpha-helicity of the interface at CC1 2/CC 3, thereby releasing CC1 3 and opening up the entire CC1 domain, culminating in the homomerization of CC1 and STIM1 activation.

Overall, the study is well-done, thorough, and increases our understanding of the mechanism of the R304W GOF mutation. The distinct gain-of-function mechanisms of R304W and the previously described L251S mutation are fascinating from a molecular perspective and indicate that STIM1 activation can be induced not only from a decrease in CC1-CC3 binding, but also from conformational alterations in the other domains of the intramolecular clamp, in particular, the CC1 2/CC 3 interface. However, the limitation of this study is that it is very narrowly focused on the R304W mutation. Previous work from this group and others has already shown important roles for CC1 homomerization in activating STIM1, and the summary cartoon in Supplementary Figure 1 is similar to those previously published by the group, at least in the critical part relating to the straightening of the kink at CC1 2/CC 3 (Fahrner JBC, 2014). The specific question of how precisely how the CC1 2/CC 3 kink opens up in the mutant, and how this would happen in the wild-type protein in which the alpha helicity is unchanged is not determined. Thus, the broader implications of this study for STIM1 function are somewhat incremental. I have the following additional comments:

We thank this reviewer for appreciating that our study has been well-done and is thorough in terms of understanding the mechanism of the R304W GOF mutation.

With respect to broader implications for physiological wt STIM1 function, in addition to the STIM1 R304W (308-310)G mutant (Fig. 7c); we have now extended our study by adding results obtained with a new STIM1 (308-310)G mutant. In contrast to R304W that increases alpha helicity and helical rigidity of the kink CC1 alpha2/CC1 alpha3, the (308-310)G mutant provides much higher flexibility to the CC1 alpha2/CC1 alpha3 linker region due a lack of side-chain at those 308-310 sites. Indeed, the STIM1 (308-310)G mutant activated Orai1 currents in a clearly store-dependent manner but at a significantly slower rate and less efficiently as wild-type STIM1, compatible with a reduced homomerization propensity as revealed by our FRET measurements (now included in Fig.7c and Supplementary. Fig. 3 and p12, l11 from the bottom).

In the discussion section (p14, l4 from the bottom to p15, l12) we have added text stating that the CC1 alpha2/CC1 alpha3 linker plays an important role in stabilizing the wild-type STIM1 quiescent state. Further, the rigidity of this linker is fine tuned for physiological signaling by enabling a STIM1 homomerization propensity and SOAR/CAD exposure precisely situated between constitutive and significantly slowed store-dependent activation. We believe that the updated model (Fig. 8) based on our new results will help our readers grasp the essence and significance of the findings.

- Figure 2. What are the effects of other key substitutions at R304? For example, Val, which is small yet highly hydrophobic (and which, according to the authors hypothesis, should also be constitutively active)? Also, Proline would be predicted to open up the CC1 2/CC 3 interface. If the authors are correct in their hypothesis that only mutations that increase alpha-helical propensity of the CC1 2/CC 3 interface should spontaneously turn on STIM1, the Pro substitution should not turn on STIM1 because even though it may open up the CC1 2/CC 3 interface, it may do little to alter the alpha-helical propensity of this region. This would distinguish between these two possibilities.

Following the reviewer's suggestion, we constructed CFP-STIM1 R304V and R304P mutants for electrophysiological analysis with co-expressed YFP-Orai1 in HEK cells. As expected, STIM1 R304V induced initial, constitutive Orai1 activation, to approximately half of the level obtained with the R304W mutation, possibly due to the smaller side chain size (now included in Fig. 2a, c; p5, 19 from the bottom). Additionally, we generated a YFP-OASF R304V-CFP conformational sensor which confirmed the expected decrease in FRET induced by extension (now included in Fig. 2d, p6, 116).

STIM1 R304P resulted in three populations with mixed activation characteristics with respect to Orai1: (i) no Ca^{2+} current activation, (ii) small store operated Ca^{2+} currents and (iii) small, constitutive Ca^{2+} currents, depicted in the Fig.1 below. Proline is a very strong helix-breaker due to an inability to maintain the backbone hydrogen bonding pattern of alpha helices and disruption of the ideal packing of atoms required for a helical geometry near the Pro. Given the cooperative nature of protein folding, it is difficult to predict and interpret the effects of a Pro substitution beyond the perturbation of the helix in which it is incorporated. In Fig. 2c and 2d, we evaluated the effects of ten different residue types at position 304 finding that hydrophobicity and size play a role, and, while we appreciate the reviewer's suggestion, given the Pro substitution yielded ambiguous data which cannot be straightforwardly interpreted in the absence of accompanying high resolution structural information, we prefer not to include the STIM1 R304P mutant data in the manuscript.

Referee 2, Fig. 1

- Supplementary figure 2 the biophysical assays here do not seem to match well with the data in Figure 2. Figure 2 shows extension of OASF in R304W and almost no extension of R304A, yet the melting temperature and SEC assays show very similar results for both mutants. Glutaldehyde cross-linking gel doesn't look very convincing a dose response curve and a negative control that does not cross-link would help. Further, given the distinct mechanisms proposed for R304W and L251S, L251S should be included in the tests shown in Supplementary Figure 2, especially panel c.

We found a decreased melting temperature for both R304W and R304A OASFext compared to WT, in isolation. This decreased melting temperature is consistent with the small, but statistically significant decrease in the FRET observed for R304A compared to WT (Fig. 2d). Further, it is important to note that the FRET experiments were performed at ambient temperature in live cells, where the differences may be less pronounced than

differences monitored nearer to the melting temperatures and where crowding effects of the components of the intracellular milieu may affect conformational changes (p10, I11).

We agree with the reviewer that the crosslinking gel is somewhat ambiguous. Therefore, to better evaluate the differences in the affinities between WT and R304W, we performed a series of new analytical ultracentrifugation (AUC) experiments using recombinant OASFext protein. Furthermore, as per the reviewer's suggestion, we also quantified the self-association affinity of the L251S mutant. Using AUC, we found that WT, L251S and R304W OASFext proteins were all in a primarily dimeric state at $\sim 0.5 \text{ mg mL}^{-1}$ (see new Supplementary Fig. 2; p11, I15). Consistent with the minimally dimeric stoichiometry, the dimer dissociation constants (K_d) were all sub- μM , suggesting high affinity monomer interactions. Estimation of the self-association constants of the dimer unit building blocks (i.e. dimer-to-tetramer association) revealed more notable differences (p 11, I17 from the bottom). The self-association affinity of the dimers for all the proteins were sub-mM, suggesting weak interactions. However, the R304W protein showed a relatively higher affinity than the WT protein (i.e. $K_d \sim 144$ versus $\sim 265 \mu\text{M}$ for R304W compared to WT, respectively; see new Supplementary Fig. 2). This apparently higher affinity is consistent with the less persistent protein bands following chemical crosslinking for the R304W protein compared to WT (Supplementary Fig. 1). The L251S protein also showed an apparently higher affinity compared to WT in the OASFext context. This increased affinity is in-line with the promoted intermolecular CC3:CC3 interactions after release of the CC1:CC3 clamp which leads to Orai1 activation (Fahrner et al., J Biol Chem 289:33231, 2014). Given the decreased CC1-CC1 interactions caused by L251S which we demonstrated in the present manuscript (Fig. 6), the higher apparent affinity of L251S OASFext compared to WT is likely driven through the promoted CC3:CC3 interactions. Indeed, we previously showed that CC3 has the highest propensity for homomeric STIM1 coiled-coil interactions using our FIRE technique (Fahrner et al., J Biol Chem 289:33231, 2014).

- Figure 4. The conclusion that Q314 or E318 are not involved in homomeric CC1-CC1 interactions is based entirely on the lack of GOF effects induced by W or K substitutions at these residues. Because it is well-known that W and K amino acids can also participate in hydrogen bonds, the conclusion that a previous report (ref 43) is incorrect is an over-interpretation of the very limited dataset here. It would be more convincing to show effects of hydrophobic substitutions such as Ala to test this idea.

As per the reviewer's suggestion, we prepared CFP-STIM1 Q314A and E318A mutants and performed electrophysiology with HEK cells co-expressing YFP-Orai1 (Fig. 2 below and p8, I5). Analysis revealed clear, store dependent Orai1 activation similar to the experiments performed with the other CFP-STIM1 mutants (Q314W, Q314K, E318W, E318K). The new additional results support our suggestion that Q314 or E318 are not involved in the gain of function mechanism of STIM1 R304W.

Referee 2, Fig. 2

Nevertheless, we do agree with the reviewer that we may be over-interpreting our Fig. 4 data, and so have added a statement (p8, I11) that more mutagenesis and additional high resolution structural information of the full cytoplasmic domain in the active and inactive conformations are required to precisely assess the

significance of the Q314 and E318 residues and their potential interactions in the STIM1 activation state.

- Figure 7b-e, the 308-310 triple Gly mutation and especially the 305-311 septuple Ala mutations are very drastic. The effects of these mutations are interpreted in the context of helix formation, but it is hard to parse out other effects in addition to helix integrity (e.g. disruption of interactions with neighboring amino acids). These mutations may not activate STIM1 through the same mechanism as R304W.

In addition to the STIM1 R304W (308-310)G mutant (Fig. 7c), we have now included data of a new STIM1 (308-310)G triple mutant in Fig. 7c, which reveals that increased flexibility in the CC1 α 2 – CC1 α 3 linker region significantly slows and less efficiently activates Orai1-mediated Ca²⁺ currents concomitant with a reduced homomerization propensity of STIM1 (308-310)G as revealed by FRET (new Supplementary Fig. 3). (p12, l11 from the bottom)

Nevertheless, we agree with the reviewer that caution is needed when interpreting either the triple or septuple mutation and have included the statement (p15, l6) that “We cannot exclude that these 308-310 triple Gly mutation and the 305-311 Ala mutations may induce additional effects beyond affecting the helix integrity, which may not occur with the R304W mutation.” Additionally, we have added in the discussion that the CC1 alpha2/CC1 alpha3 linker plays an important role in stabilizing the wild-type STIM1 quiescent state. Further, the rigidity of this linker is fine tuned for physiological signaling by enabling a STIM1 homomerization propensity and SOAR/CAD exposure precisely situated between constitutive and significantly slowed store-dependent activation (p15, l8).

Reviewer #3 (Remarks to the Author):

This is an interesting manuscript reporting on the role of the C1 coiled-coil domains interaction in STIM1 and how the interaction is regulated by the hydrophilic environment of the C1a1 domain. The authors used the Stormorken mutation that causes gain-of-function of STIM1 to report how the mutation activates STIM1 and when doing so, discovered an important function of C1a1 in the activation of STIM1 and Ca²⁺ influx by the Orai1 channel. The findings are a major advance in our understanding of STIM1 activation and thus activation of Ca²⁺ influx and Ca²⁺-related cell functions and pathologies. The experiments are well done and provide ample structural (NMR), biochemical (FRET and CD) and electrophysiological (current) evidence to support the main conclusion.

I have only minor comments that relate to style of writing, the use of some terms and in particular the discussion. Although the manuscript is well written and the writing is mostly straight forward, the discussion does not discuss in any length the novel finding from structure-function point of view. The results section is written very well from the perspective of the significance of the R304W mutation and the hydrophilicity of C1a1 on STIM1 unfold and channel activation, this is barely mentioned in the discussion, let alone discussing whether and how the half turn extension of this domain by changing the hydrophilic environment can occur as a result of store depletion. i.e. how store depletion is transmitted to C1a1 to cause the extension, stabilize homomerization and activate STIM1. It seems that this should be the bulk of the discussion since these are the novel findings and as far as the disease is concern GOF is GOF.

We thank the referee for her/his very good review.

We included new STIM1 (308-310)G data in Fig. 7, which revealed that higher flexibility in this region affect STIM1/Orai1 activation. Indeed, the STIM1 (308-310)G mutant activated Orai1 currents in a clearly store-dependent manner but at a significantly slower rate and less efficiently as wild-type STIM1, compatible with a reduced homomerization propensity as revealed by our FRET measurements (now included in Fig.7c and Supplementary. Fig. 3; p12, l11 from the bottom). In conclusion, these experiments are perfectly in line with the hypothesis that flexibility / rigidity of the CC1 α 2 – CC1 α 3 linker represent a critical component for STIM1 activation. We have integrated the new data in the discussion text, emphasizing the role of the CC1 α 2 – CC1 α 3 linker in stabilizing the STIM1 quiescent state. Further, the rigidity of this linker is fine tuned for physiological signaling by enabling a STIM1 homomerization propensity and SOAR/CAD exposure precisely situated between constitutive and significantly slowed store-dependent activation (p15, l8).

Figure 2b, 3b: The behavior of R304K is interesting. Stabilization of the quiescent state may also mean that more store depletion and dissociation from STIM1 is required for its activation. This can be tested by determining if the STIM1(D76A) or other mutation that affect STIM1 Ca²⁺ binding can facilitate activation of this mutant to the same extent as wild-type. The same question is applicable to L300W and its very slow activation. The latter should drive further the main novel and important message that C1a1 controls homomerization of STIM1.

We have prepared the constructs CFP-STIM1 R304K D76A and L300W D76A, respectively, and performed electrophysiology with co-expressed YFP-Orai1 in HEK cells. Both double mutants showed constitutive Ca²⁺ current due to the D76A induced insensitiveness of the ER-luminal EF-hand, simulating Ca²⁺ depleted store conditions. There was, however, no marked difference in the rate as well as extent of Ca²⁺ current activation visible, when comparing wild-type STIM1 D76A with STIM1 R304K D76A or STIM1 L300W D76A (see Fig. 1 below).

Referee 3, Fig. 1

Therefore, we would prefer not to include this Figure but, we have additionally mentioned in the manuscript that the slower onset of store-dependent activation observed with the STIM1 R304K or L300W mutants suggested that these residues in the CC1 alpha2 region also contributed to stabilization of the quiescent state of STIM1 (p5, l4 from the bottom; p7, l4).

Figure 4c: the significance of the effect of E318K is important and should be discussed here when describing first the mutant rather than mentioning it later in a glance.

The results of Fig. 4 (STIM1 Q314 and E318 mutations) have been mentioned in the results section (p8, l3), in as that these mutations showed some effect on the rate of activation but all of these mutants tested exhibited clear store-dependent rather than constitutively activated Ca²⁺ currents.

We have additionally stated "In summary, our data suggested it as unlikely that an R304W-mediated disruption of R304:Q314/E318 interactions led to constitutive STIM1/Orai1 activation, however, additional high resolution structural information of the full cytoplasmic domain in the active and inactive conformations is required to precisely assess the significance of the Q314 and E318 residues and their potential interactions in the STIM1 activation state."(p8, l10).

In page 9 and later the authors suggest that R304 both stabilize homomerization and disrupts the C1-C3 clump while that data in Figure 5 provides no evidence for clump disruption. Can the authors clarify why they think or what evidence support clump disruption.

This is one of the main result that distinguishes the GoF mutant STIM R304W from STIM1 L251S. While both mutants assume the extended conformation as shown by the respective OASF conformational sensor, the mechanism is distinctly different. The CC1-CC3 clamp is disrupted by the L251S mutation, as derived from fragment interactions examined by the FIRE approach, whereas CC1 R304W still showed an interaction with CC3. Although this interaction is seen with fragments, it can be apparently not formed in the OASF entity or the whole STIM1, as they adopt the extended, constitutively active conformation. Mechanistically, we suggested the increased rigidity in the CC1 alpha2/CC1 alpha3 linker interfered with the formation of the inhibitory CC1-CC3 clamp together with an increased homomerization of the CC1 domain, both effects caused by the R304W mutation (p8, l6 from the bottom till p9, l5 from the bottom).

Figure 7: The control of (308-310)G alone on the current is missing.

Currents obtained with co-expression of STIM1 (308-310)G together with Orai1 have been added to Fig 7c. Please refer to our first statement, where STIM1 (308-310)G is described, in particular with reference to

mechanistic aspects of wild-type STIM1 activation..

Some of the unclear sentences/words:

Page 5, second paragraph, line 5 should read: noteworthy mentioning that W has/contain

Page 8, lines 18-290: Sentence not clear and what enough degrees of freedom means in this context

Page 9, line 5 from bottom: add To before STIM1

Page 10, line 9: drive is not accurate, cause is better

Page 11, line9-10: should read the predicted α 2helix was extended, thus

Page 12, line 4: Remove suffer of; line 17: change correlation to to association with; line 22, add upon after speculated

Page 13, line9, add the afterin

We have clarified and corrected all these points.

Reviewers' Comments:

Reviewer #1:

Remarks to the Author:

The authors have fully addressed my concerns in the revised version of the manuscript.

Reviewer #2:

Remarks to the Author:

The authors have done a reasonable job of elucidating the effects of the R304W mutation, which activates STIM1 by releasing the intramolecular clamp in the CC1 domain of STIM1. Precisely how this clamp (mediated by the CC1 α 2/CC α 3 kink) is released by the mutation, is however, not addressed and the revision does not advance the underlying mechanism. More importantly, how the CC1 α 2/CC α 3 would straighten in the wild-type protein in which the alpha helicity of R304 is unchanged is not determined. Because previous studies (including this group) have already hypothesized that release of the clamp would have to involve CC1 homomerization and unraveling of the kink at the CC1 α 2/CC α 3 region (encompassing R304), the effects of R304W in inducing these changes is not entirely surprising. Thus, the study is not so much about STIM1 works, but rather a phenotypic characterization of the various effects induced by the R304W activating mutation.

Reviewer #3:

Remarks to the Author:

The authors carefully and completely addressed all my concerns. I have no further comments and the manuscript is now ready for publication.

Reviewer #2 (Remarks to the Author):

The authors have done a reasonable job of elucidating the effects of the R304W mutation, which activates STIM1 by releasing the intramolecular clamp in the CC1 domain of STIM1. Precisely how this clamp (mediated by the CC1 2/CC 3 kink) is released by the mutation, is however, not addressed and the revision does not advance the underlying mechanism. More importantly, how the CC1 2/CC 3 would straighten in the wild-type protein in which the alpha helicity of R304 is unchanged is not determined. Because previous studies (including this group) have already hypothesized that release of the clamp would have to involve CC1 homomerization and unraveling of the kink at the CC1 2/CC 3 region (encompassing R304), the effects of R304W in inducing these changes is not entirely surprising. Thus, the study is not so much about STIM1 works, but rather a phenotypic characterization of the various effects induced by the R304W activating mutation.

Answer to Reviewer #2:

With all due respect, we partially disagree with the reviewer's view in as that the effects of R304 mutation were not entirely surprising. While the role of the CC1 alpha1 and alpha3 within STIM1 have been already characterized, it arose quite unexpectedly that R304 within CC1 alpha 2 is a key site, the mutation of which to R304W elicited a dual mechanism affecting both CC1 homomerization and the linker region between CC1 alpha 2 and CC1 alpha 3. In particular the flexibility (or rigidity) of the latter region within wild-type STIM1 is apparently fine tuned for physiological signaling by enabling a STIM1 homomerization propensity and SOAR/CAD exposure precisely situated between constitutive and significantly slowed store-dependent activation.